# Assessment of Recombinant β-Propeller Phytase of the *Bacillus* Species Expressed Intracellularly in *Yarrowia lipolityca*

**DOI:** 10.3390/jof11030186

**Published:** 2025-02-26

**Authors:** Liliya G. Maloshenok, Yulia S. Panina, Sergey A. Bruskin, Victoria V. Zherdeva, Natalya N. Gessler, Alena V. Rozumiy, Egor V. Antonov, Yulia I. Deryabina, Elena P. Isakova

**Affiliations:** 1Vavilov Institute of General Genetics, Russian Academy of Sciences, 119991 Moscow, Russia; maloshenoklg@gmail.com (L.G.M.); jpanina@yandex.ru (Y.S.P.); brouskin@vigg.ru (S.A.B.); 2A.N. Bach Institute of Biochemistry, Research Center of Biotechnology of the Russian Academy of Sciences, Leninsky Ave. 33/2, 119071 Moscow, Russia; vjerdeva@inbi.ras.ru (V.V.Z.); gessler51@mail.ru (N.N.G.); rozumiy01@mail.ru (A.V.R.); zhora-antonov@list.ru (E.V.A.); yul_der@mail.ru (Y.I.D.)

**Keywords:** β-propeller phytase, yeast, *Yarrowia lipolytica*, *Bacillus* species, refolding

## Abstract

Phytases of the PhyD class according to their pH optimum (7.0–7.8) and high thermal stability can claim to be used in the production of feed supplements. However, today they have no practical application in feed production because there are no suitable producers sufficient for its biotechnological production compared to the PhyA and PhyC class ones. Moreover, in most cases, the technologies with the enzymes produced in secretory form are preferable for the production of phytases, though upon microencapsulation in yeast-producing cells, the phytase thermal stability increases significantly compared to the extracellular form, which improves its compatibility with spray drying technology. In this study, we assayed the intracellular heterologous expression of PhyD phytase from *Bacillus* species in the *Yarrowia lipolytica* yeast cells. While the technology has been successfully used to synthesize PhyC phytase from *Obesumbacterium proteus*, PhyD phytase tends to aggregate upon intracellular accumulation. Furthermore, we evaluated the prospects for the production of encapsulated phytase of the PhyD class of high enzymatic activity when it accumulates in the cell cytoplasm of the *Y. lipolytica* extremophile yeast, a highly effective platform for the production of recombinant proteins.

## 1. Introduction

Enzyme application in animal feeding is considered an indispensable technological tool for using available feedstock effectively, increasing weight gain, and reducing the cost of mineral additives [1,2,3]. Phytases (esterases hydrolyzing 1,2,3,4,5,6-myoinositol hexaphosphate) are one of the most valuable feed enzymes [3,4]. Phytate (myo-inositol 1,2,3,4,5,6-hexakis dihydrophosphate), being the main source of inositol and phosphorus (P_i_) storage, usually reaches 60–90% of the total P_i_ in oilseeds, cereals, and legumes [5] and causes deficiencies in essential dietary minerals, namely Fe^2+^, Zn^2+^, Mg^2+^, and Ca^2+^ acting as an anti-nutritional factor [3,4]. Phytate cannot hydrolyze in the gastrointestinal tract of pigs, poultry, fish, and other monogastric productive animals [6], which requires phytase additives in feed premixes. Based on the catalytic mechanism, three-dimensional structure, and specific properties of the sequence, phytases are divided into four classes, namely histidine acid phosphatases (HAPhys), cysteine phytases (CPhys), purple acid phosphatases (PAPhys), and β-propeller phytases (BPPhys) [7]. Among them, BPPhys (with the structure of a beta-propeller with six blades) are considered the most widespread. They are assumed to play a leading role in the circulation of phytate-P in soil and water [8]. BPPhys form the only class of phytases with an activity at a neutral and alkaline ambient pH [9]. Moreover, BPPhys show better thermal stability, proteolytic stability, and absolute substrate specificity, a good replacement for commercial phytases [7]. Phytases are also classified based on the sites where the hydrolysis of the inositol ring begins into 3-phytases (EC 3.1.3.8), 5-phytases (EC 3.1.3.72), and 4/6-phytases (EC 3.1.3.26). As a rule, 3-phytases are characterized by the initiation of dephosphorylation on the third carbon atom of the inositol ring, whereas 6-phytases can initiate the cleavage of phytate on the sixth carbon atom, which can lead to an increase in dephosphorylation of the inositol ring. Although 6-phytases have a higher efficiency of phytic acid dephosphorylation and improve growth features compared to 3-phytases [10], no one phytase could completely dephosphorylate it. Structurally and enzymatically, phytases from various kingdoms of the living world are also divided into four classes. The enzymes of the PhyA and PhyC classes have been used in feed production [11]. The PhyA class is found mainly in deuteromycete fungi, in particular, in *Aspergillus niger*, *Aspergillus fumigatus*, *Aspergillus ficuum*, and *Penicillium oxalicum*, which are produced in secretory form [12,13]. The enzymes are highly glycosylated and possess high thermal stability and two pH optima (on average, a pH of 2.5 and a pH of 5.5). As for substrate specificity, they belong to 3-phytases, i.e., they cleave phosphodiester bonds in 1-, 3-, and 5-positions of the myoinositol ring at a high rate [14]. Phytases of the PhyC class are found in bacteria of the *Escherichia coli* group and the *Bacillus* genus [15,16].

Unlike the PhyA class, the enzymes show a single pH optimum within 5.0–6.0, with the thermal stability of 55–60 °C with a substrate specificity of 6-phytases, i.e., they have no preference for the cleavage of any phosphodiester bond [17]. It should be noted that PhyC phytases from *Bacillus* spp. are the β-propeller ones with an optimal pH from 6.0 to 9.0, suitable for the neutral gastrointestinal tracts of animals such as trout and cyprinids. They exhibit a high thermal stability [15,16]. The practical application of phytases of the PhyC class is complicated because they are not secreted in vivo but accumulate in the periplasm or lysosomes. Therefore, the design of secreting producers of the enzymes needs recombinant technologies, where the methylotrophic yeast of *Pichia pastoris* is most popular. The technological disadvantage of PhyC phytases compared to the PhyA ones is their lower thermal stability, which inhibits their activity upon using a spray dryer, where the granulation process undergoes at 70–80 °C [18]. However, phytases of the PhyC class possess significantly higher specific activity in the gastrointestinal tract of productive animals compared to that of the PhyA type. Due to the lower substrate specificity to phosphor–ester bonds in various positions of the myoinositol ring, they provide a complete release of phosphorus from the grain phytate. Therefore, recently, the interest of the manufacturers and phytase consumers of the PhyC class has been steadily growing. Recent studies have presented data on a new type of consensus phytase class G. PhyG (PHY-13594) was obtained from the fermentation of the *Trichoderma reesei* strain, which expressed the consensus bacterial gene of the 6-phytase with high activity within a wider pH than the PhyB does.

The new PhyG showed a high activity at a low pH of 1.5–2.0 indicating that PhyG can rapidly hydrolyze phytate and then reduce the negative effect of phytate on nutrient availability [19]. Phytases of the PhyB and PhyD classes have no practical application in feed production since at the moment there are no strain-producers convenient enough for biotechnological production, which provide the complete release of phosphorus from grain phytate. Meanwhile, phytases of the PhyD class of microbial (bacillary) and plant origin, according to their pH optimum (7.0–7.8) and high thermal stability, can be applied in feed production. Before, we proposed an original technology for the production of a phytase of the PhyC class from *O. proteus*. It hypothesizes the rejection of the production of the enzyme in a secretory form, i.e., the recombinant product accumulates in the cytoplasm of the extremophilic *Y. lipolytica* yeast. Although the producer shows a lower biomass yield per one liter of culture medium than those of the recombinant strains based on *Y. lipolytica*, methylotrophic yeast, and fungi, it has some technological advantages. The main one is the possibility for bioremediation of waste from slaughterhouses and the oil extraction industry to form a protein–enzyme feed additive. Thus, waste and effluents are cleaned of hydroperoxides and xenobiotics, the technology need not separate the biomass from the cultural medium, which significantly reduces both the cost of drying the supplements and the waste generation of biotechnological production. Due to microencapsulation in yeast-producing cells, the thermal stability of phytase significantly increases compared to that of extracellular form that improves its compatibility with premix spray-drying technology. Due to microencapsulation of the phytase, unproductive enzyme loss because of acid denaturation and proteolysis upon passing the stomach of livestock and poultry are significantly decreased [4]. Currently, the bacteria of the *Bacillus* genus are important objects for phytase production. Their phytases have been studied thoroughly because they possess some unique features. Moreover, there is a possibility of their production for animal feeding [20]. *B. subtilis* belongs to a species of Gram-positive, rod-shaped, spore-forming bacteria, which can produce many secondary metabolites, including subtilin, α-amylase, phytase, and nattokinase [21,22,23,24]. It is a non-pathogenic organism. Numerous genes of *B. subtilis* phytase have been successfully cloned in some bacterial species, namely, *E. coli* for its extracellular secretion [18,25].

In the presented study, we demonstrate the features of novel active isoforms of phytases from the bacteria of the *Bacillus genus*, capable of accumulating in the *Y. lipolytica* cells, undergoing aggregation and subsequent refolding using osmolytes.

## 2. Materials and Methods

### 2.1. Strains and Growth Conditions

*E.coli* XL-1Blue strain (recA1 endA1 gyrA96 thi-1 hsdR17 supE44 relA1 lac [F proAB lacIqZΔM15 Tn10 (Tetr)]) was used for standard genetic engineering study (plasmid construction, plasmid DNA production). The culture was grown at 37 °C in LB, g/L medium: trypton—10; yeast extract—5; NaCl—5; if necessary, ampicillin (Sigma, St. Louis, MI, USA) was added at a concentration of 100 µg/mL to raise transformants. Bacterial strains were used, namely *Bacillus subtilis* UQM 41285 (ATCC 23857, strain 168), *Bacillus cereus* ATCC 11778, *Bacillus licheniformis var. mycoides* 537, and *Bacillus amyloliquefaciens* (B10986) (VKPM). *Y. lipolytica PO1f* cells (MatA, leu2-270, ura3-302, xpr2-322, asp-2) were deposited in the CRM-Levres collection (France, number CLIB-7240).

### 2.2. DNA Extraction

High-quality total genomic DNA was isolated using the ExtractDNA Blood and Cells kit (BC111M, Total DNA extraction kit for whole blood, animal cells, and bacteria, Evrogen, Moscow, Russia) according to the manufacturer’s protocol. The quality and quantity of the isolated DNA were tested using electrophoresis in 1.2% agarose gel stained with ethidium bromide and using a Nanodrop ND-1000 spectrophotometer (Nanodrop Technologies, Wilmington, DE, USA) at wavelengths of 230, 260, and 280 nm (Appendix A).

### 2.3. In Silico Bioinformatics Analysis

The initial step involved searching for nucleotide and amino acid sequences of phytases in annotated *Bacillus* strain genomes available in the NCBI GenBank database (https://www.ncbi.nlm.nih.gov/, accessed on 11 March 2023). Reference sequences from studies [15,26], along with keyword searches (“Phytase… Bacillus”, “beta-propeller phytase Bacillus”), were used. Orthologs of PhyD-class phytase genes were identified using NCBI NucleotideBLAST and ProteinBLAST tools (https://blast.ncbi.nlm.nih.gov/Blast.cgi, accessed on 11 March 2023). All retrieved sequences (Appendix A) were analyzed via multiple sequence alignment using Clustal W [27] in BioEdit program (version 6.0.7), followed by classification into isoforms exhibiting high amino acid residue homology (Appendix A). Universal primers were designed for each isoform (Appendix A) to amplify phytase genes from the following *Bacillus* strains: *B. subtilis* UQM 41285 (ATCC 23857, strain 168), *B. cereus* ATCC 11778, *B. licheniformis var. mycoides* 537, and *B. amyloliquefaciens* (B10986, VKPM). Sequence alignments were generated with Clustal W [27] in BioEdit program (version 6.0.7). Clustal W parameters included sequence weighting, position-specific gap penalties, and weight matrix selection to enhance alignment sensitivity [27].

### 2.4. Amplification of Genes Encoding Phytases

The primers were designed based on nucleotide sequences encoding phytases published in GenBank.

PCR was performed using pairs of primers listed in Appendix A and a high-precision thermostable polymerase Q5 with 3→5 exonuclease activity and fused with a DNA-binding Sso7d domain that enhances processivity. The reaction mixture was prepared according to the manufacturer’s protocol. To amplify the genes encoding phytases, the pairs of primers and genomic DNA isolated from six species of the *Bacillus* bacteria were used.

All the PCR for sequencing was performed under the following conditions: 1 cycle (30 s 98 °C), then, 35 cycles (10 s 98 °C, 20 s 56 °C, 40 s 72 °C), and 1 cycle (5 min 72 °C).

### 2.5. Isolation of PCR Products from the Gel

PCR products were isolated from the gel using the QIAquick Gel Extraction Kit (Qiagen, Hilden, Germany), according to the manufacturer’s protocol.

### 2.6. Construction of an Integrative Expression Vector tphyD-BS-1 or tphyD-BS-2 Phytase Gene

Cloning of the phyD-BS-1 or phyD-BS-2 phytase gene was performed with PCR, using a nucleotide sequence previously amplified on genomic DNA as a matrix. The use of a pair of Pr-Phy-Bs/168/siamensis-f_BamHI and Pr-Phy-Bs/168/siamensis_NotI primers (Appendix A, Pair 8) allowed a fragment to be cut off encoding a region of 26 codons long corresponding to the secretory leader. As a result, phytase loses the ability to secrete into the environment and accumulates in the cytoplasm. The primer of Pr-Phy-Bs/168/siamensis-f_BamHI provided the introduction of the BamHI restriction site, and the introduction of an artificial GGA (Gly) codon at the N-terminus of the modified tPhyD-Bs-1 gene, which, being just after the initiator codon ATG according to the Varshavsky rule [28], increases the cytoplasmic stability of the product in the yeast. Thus, the phyD-Bs-1 gene was obtained, flanked by the restriction sites of BamHI and NotI. According to the results of electrophoresis in agarose gel, the size of the PCR product corresponded to the expected one of 1100 bp. The PCR product was purified using the QIAquick Gel Extraction Kit (Qiagen, Hilden, Germany), treated with BamHI and NotI restrictases, and ligated with the linearized pUV-LT3 vector [29] treated with BamHI and NotI restrictases. Ligation was performed in a 1.5 mL test tube as follows, the vector and the gene were mixed in a ratio of 1:4, a buffer for ligase (Fermentas, Waltham, MA, USA), DNA ligase of phage T4 (Fermentas, Waltham, MA, USA—5 units), added ultrapure water to 20 µL and incubated at room temperature (RT) for an hour. The ligase mixture was used to transform the *E. coli* XL1Blue strain according to a standard protocol using Ca^2+^ ions. The transformed *E. coli* cells were grown in a solid LB medium with kanamycin (50 µg /mL). Ten selected clones were raised in a liquid LB medium with kanamycin (50 µg/mL), and plasmid DNA was isolated, which was checked for the desired insert of 1100 bp using PCR. The absence of substitutions in the target sequence of the obtained genes and the correctness of the assembly of the genetically engineered structure were confirmed with sequencing using the Sanger method. The resulting structures were named pUV3-PhyD-Bs-1 and pUV3-PhyD-Bs-2 (Appendix A).

### 2.7. Transformation of the Y. lipolytica Yeast Using a Plasmid Structure with the Phytase Gene

The introduction of the pUV3-PhyD-Bs-1 DNA construct into the *Y. lipolytica PO1f* cells (MatA, leu2-270, ura3-302, xpr2-322, axp-2) (deposited in the CIRM-Levures collection (France, the number CLIB-724) was performed with the transformation method using Li^+^ salts, as described before [30]. The selection of the transformants was performed on a minimal synthetic medium free of uracil with the addition of leucine at 28 °C.

### 2.8. Genomic DNA Extraction

Chromosomal DNA was isolated from the *Y. lipolytica PO1f* transformants using the yeast kit for isolating chromosomal DNA from yeast (diaGene, Moscow, Russia) according to the manufacturer’s protocol. The YeastKit kit is designed to isolate quickly chromosomal DNA from small amounts of yeast biomass and it permits obtaining the chromosomal DNA suitable for the production of the PCR fragments, which are at least 10,000 bp long. The kit includes a special suspension for cell lysis and glass beads.

### 2.9. Selection of Transformants with PCR

The transformants were selected using minimal synthetic medium free of uracil, with leucine at +28 °C. The grown colonies (20–50 clones) were ordered by transferring to solid YNB medium with 1% glucose and leucine. The insertion of the desired size of 1100 bp in the clones was confirmed with PCR using primers Pr-Phy-Bs/168/siamensis-f_BamHI and Pr-Phy-Bs/168/siamensis_NotI and high-precision thermostable polymerase Q5 as described before. To assay the productivity of transformants of *Y. lipolytica PO1f* (pUV3—PhyD-Bs-1 or pUV3—PhyD-Bs-2) ten clones were selected, where the PCR method using another pair of primers Pr-Vdac-Forw and M13 fwd (after the terminator) reconfirmed the chromosomal integration of a fragment containing *URA3*, VDAC-phyD-BS-1 (phyD-BS-2)-Txpr. In all the selected transformants, there was an 1800 bp long fragment.

### 2.10. RNA Isolation and Reverse Transcription Reaction

Total RNA from *Y. lipolytica* biomass was extracted using the RNeasy Mini Kit (Qiagen, Hilden, Germany), and 2.5 μg of each sample was treated with DNase (Ambion, Life Technologies, Carlsbad, CA, USA). 3 mL of yeast culture was centrifuged, the pellet was to mechanical disruption with liquid nitrogen and was resuspended in buffer RLC (RNeasy Mini Kit, Qiagen, Hilden, Germany) and further all procedures were performed according to the manufacturer’s protocol. The concentration and purity of the isolated RNA was determined using a UV spectrophotometer NanoDrop ND-1000 (Thermo Fisher Scientific, Wilmington, DE, USA). The OD260/OD280 values of the RNA samples, reflecting their average purity, ranged between 1.9 and 2.1. Furthermore, the integrity of the RNA isolates was verified through agarose gel electrophoresis (Appendix A), according to a standard method [31].

### 2.11. Quantitative Real-Time Reverse Transcriptase Polymerase Chain Reaction (qRT-PCR)

cDNA templates for all quantitative real-time reverse transcriptase polymerase chain reactions (qRT-PCR) were synthesized from 2500 ng of total RNA using the MMLV RT kit (Evrogen, Moscow, Russia) following the manufacturer’s protocol. RNA concentration was determined using a fluorometer (Qubit™ fluorometric instrument, Invitrogen Q32857, Carlsbad, CA, USA). Gene-specific primer for isoform phytase tPhyD-Bs (Fw: GAAGGACTGACAATCTATTATG, Rw: AACCGAGAACATCAATACC) were designed using software Beacon Designer (version 4.0) for SYBR® Green assays and pair primers for the Act1 (Fw: CGAGCGAATGCACAAGGA, Rv: GCGGTGATCTTGACCTTGATG) were taken from [32]. In all analyzed samples, the value of the threshold cycle for ACT1 was the same (Appendix A) The qRT-PCR experiments were performed on an Eco Real-Time PCR System (Illumina, San Diego, CA, USA). Amplification efficiency for each primer set was calculated by serially diluting the exponential-phase cDNA template. Melting curve analysis was performed for each pair of primers after each run in an Eco Real-Time PCR System instrument to confirm the specificity of the primers. DNA contamination was checked by (no reverse transcription) PCR for each RNA sample in Eco Real-Time PCR System (Illumina, San Diego, CA, USA) using target primers (primer pair Pr-Phy-Bs/168/siamensis-f_BamHI and Pr-Phy-Bs/168/siamensis_NotI). No amplification was seen after 40 cycles of amplification (Appendix A). The specificity of each primer pair was verified via electrophoresis on a 1.2% agarose gel followed by ethidium bromide staining. The PCR products were directly purified from the reaction mixture using the Cleanup S-Cap kit (Evrogen, Moscow, Russia) and sequenced using the Sanger method. Only one band was observed in the gel. All qRT-PCR experiments were performed on an Eco Real-Time PCR System (Illumina, San Diego, CA, USA). The amplification reaction was carried out using a ready-made mixture of PCR qPCRmix-HS SYBR+ROX (Evrogen, Moscow, Russia) according to the manufacturer’s instructions. Amplification program: 5 min at 95 °C, 45 cycles (95 °C 15 s, 58 °C 20 s, 72 °C 30 s), and melting reaction product (Melting Curve program: 70–95 °C, +1 °C/s). The mRNA level was normalized to the expression of Act1 (Appendix A), which was constantly expressed under all experimental conditions. The qRT-PCR results were elaborated using the 2−ΔCT method [33]. All experiments were carried out in 3–4 replicates. Four independent experiments were conducted; the paper presents the results of the most representative experiment. Statistical analyses were performed using Microsoft Excel software. Statistical significance between experimental groups was determined using the Student’s *t*-test (confidence interval 95%) [34].

### 2.12. Enzyme Activity Quantification

Each clone was grown on a minimum YNB medium enriched with 2% glucose and leucine and was transferred to a liquid medium, being a 5% aqua suspension of meat and bone meal, pH 5.3. The clones were cultured for 34 h at 28 °C. The biomass, with the insoluble medium fraction, was collected with centrifugation, washed with saline solution, and reprecipitated. The resulting precipitate was used to obtain the homogenates for assaying the phytase activity.

### 2.13. Phytase Expression (Cultivation)

The transformed *Y. lipolytica* yeast was cultivated on the solid cultural medium containing (g/L) yeast extract “DIFCO”—2.5; bactopepton—5.0; agar—20.0; glycerin—7.5; malt extract—3; pH 5.5. The liquid medium YPD for cultivating transformants contained (g/L) yeast extract—10.0; glucose—10.0; peptone—20.0; solid medium YPD additionally contained agar—20.0 g/L. Minimum medium with plant substrates contained (g/L) MgSO_4_ × 7H_2_O—0.5; NaCl—0.1; CaCl_2_—0.05; (NH_4_)_2_SO_4_—0.3; glucose—10.0; vegetable substrate—2.0. The solid medium for the spot test contained tris-acetate buffer (200 mM)—25 mL; agar-agar—0.25 g; calcium chloride (5%)—2.5 mL; sodium phytate (Sigma Aldrich, St. Louis, Missouri, USA)—0.5 g. Sodium phytate and calcium chloride solution were added after sterilization to the ready-cooled tris-agar medium. The *Y. lipolytica W29* and *Y. lipolytica Po1f* (pUV3-Op) yeast was grown in Erlenmeyer flasks on a minimal semi-synthetic medium with glycerol (0.6–1%) as the main carbon source at pH 5.5, as described in [35]. Yeast biomass was raised at the stationary growth phase (24 h).

To perform refolding, protein aggregates (inclusion bodies) were isolated from phytase transformant samples grown in Erlenmeyer flasks at 29 °C and stirring at 150 rpm. for 72 h on a medium containing peptone 2%, yeast extract 1%, glycerol 1%; a pH of 5.5 according to the protocol described in [36] with some modifications. Upon isolation of the inclusion bodies, the phosphate–salt buffer was substituted with 50 mM tris-HC l buffer; a pH of 7.0.

### 2.14. Preparation of Cellular Homogenate

The cellular homogenate was obtained as described before with some modifications [37]. The yeast cells were disrupted at 0 °C with an ultrasonic disintegrator QsonicaQ500 (Farmacia, Stockholm, Sweden) using eight pulses for 90 s interrupted by cooling periods every 30 s.

### 2.15. Assay of Phytase Activity in the Y. lipolytica Transformants

To obtain an enzyme preparation, the destroyed biomass was mixed with a Na-acetate buffer (pH 6.2) in a ratio of 1:5 and centrifuged at 12,000 rpm for 20 min. A supernatant was used for research. Phytase activity was determined spectrophotometrically at 700 nm by the formation of molybdenum blue as described in [38]. The phytase activity unit (FYT) is calculated as the amount of the enzyme releasing 1 µmol of the phosphate ion per minute under the above-mentioned conditions.

### 2.16. Protein aggregation

*Protein aggregation* was monitored by measuring turbidity at 340 nm. Aggregates purified from DNA and some yeast protein impurities that were paralleled for refolding were dissolved in 5 mL of 50 mM Tris-HCl buffer, pH 7.0, analyzing the amount of protein according to Bradford. Purified inclusion cells were solubilized with 8 M urea and placed 200 µL per well of a 96-well plate (Corning Inc., New York, NY, USA). Changes in turbidity were recorded at 340 nm using a Synergy HTX tablet reader (BioTek Instruments, Inc., Charlotte, VT, USA). Three biological (n > 3) and three technical (n > 3) repeats were performed for each experiment.

### 2.17. Phytase Refolding

Phytase refolding from purified inclusions was performed according to [39]. The aggregates purified from DNA and some impurities of yeast proteins were solubilized in 5 mL of 50 mM Tris-HCl buffer, a pH of 7.0; containing 8 M urea, assaying the protein amount according to Bradford [40]. The denatured protein was diluted to a concentration of 0.1 mg/mL in a refolding buffer containing 100 mM tris-HCl, a pH of 7.0; 0.5 mM CaCl_2_, 2 M proline, at 10 °C for four days. Phytase activity was assayed in 1 microgram of protein in a sample (10 µL) at a pH of 6.0, pH of 7.0, and pH of 8.0 using molybdenum blue as described in [38]. Refolding was assessed after treatment of purified protein aggregates in 8M urea and 100 mM Tris buffer; pH of 7.0.

### 2.18. Electrophoresis

Native electrophoresis was performed on the plates measuring 11 × 11 cm in a gradient polyacrylamide gel (5–20% polyacrylamide) in a tris-HCl buffer, pH of 8.8, as described in [41].

### 2.19. Assay of the Protein Amount

Total protein was assayed by the Biuret method and Bradford one with BSA as a standard. The optical density of solutions was assayed on a spectrophotometer at 595 nm [40].

### 2.20. Structural Characteristics of PhyC and PhyD Phytases

Predicted N-glycosylation sites were calculated using bioinformatic service DTU Health Tech NETNGLYC-1.0 (https://services.healthtech.dtu.dk/services/NetNGlyc-1.0/, accessed on 1 June 2023).

Predicted O-glycosylation sites were calculated using the bioinformatic services DTU Health Tech NetOGlyc 4.0 (https://services.healthtech.dtu.dk/services/NetOGlyc-4.0/, accessed on 1 June 2023).

GRAVY was calculated using ProtParam the protein identification and analysis tool on the Expasy Server (https://web.expasy.org/protparam/, accessed on 1 June 2023). The average hydropathy (GRAVY) of a linear polypeptide sequence is calculated by summing the hydropathy values of all amino acids and dividing by the number of residues in the sequence. An increase in the positive score indicates greater hydrophobicity, but it does not account for how the protein folds in three dimensions, or the percentage of residues buried in the protein’s hydrophobic core.

Number of Cys и total disulfide bonds were calculated using the software facilitie to analyse NPS@’s data: AnTheProt and MPSA (https://npsa-prabi.ibcp.fr/cgi-bin/npsa_automat.pl?page=/NPSA/npsa_cysteines.html, accessed on 1 June 2023).

## 3. Results

### 3.1. In Silico Search for PhyD Phytase Genes (Comparative Genome Analysis)

In this study, we searched for genes encoding PhyD phytases in the genomes of *Bacillus* species available in the GenBank NCBI database (https://www.ncbi.nlm.nih.gov/, accessed on 11 March 2023). We included species such as *Bacillus subtilis*, *Bacillus cereus*, *Bacillus licheniformis*, *Bacillus siamensis*, *Bacillus amyloliquefaciens*, and *Bacillus spizizenii*. As a result of bioinformatics (comparative) analysis, we identified five variants of phytases, which differ significantly in their amino acid composition but are capable of hydrolyzing phytate in the manner of PhyD phytases. The final product of this hydrolysis is inositol trisphosphate Ins(2,4,6)P3 (Figure 1) [42].

According to the data presented in Figure 2, one of the phytase isoforms (Figure 1A) is found in the genomes of four out of the six *Bacillus* species of interest (hereafter referred to as PhyD-isoform-1). A second isoform (Figure 1B) occurs in two species (hereafter PhyD-isoform-2), while a third isoform (Figure 1C) is found only in *B. licheniformis* (hereafter PhyD-isoform-3). The fourth (Figure 1D) and fifth isoforms (Figure 1E) occur exclusively in *B. cereus* (hereafter PhyD-isoform-4 and PhyD-isoform-5, respectively). The variability in these phytase isoforms correlates with the phylogenetic clustering of the *Bacillus* species, as shown in the paper by Khurana et al. [43], which employed pairwise distance values between 178 *Bacillus* genomes. The phylogenetic tree was constructed using hierarchical clustering with the Multiple Experiment Viewer (MeV) software (version 4.9.0) [44], and the dendrogram was generated with the interactive tree of life (iTOL) tool (http://itol.embl.de, accessed on 1 March 2023) [45]. The *Bacillus* species were divided into 11 clusters based on monophyletic clustering, with *B. subtilis* in cluster D, *B. siamensis*, *B. amyloliquefaciens*, and *B. siamensis* in cluster E, and *B. cereus* in cluster G. Phylogenetic analysis of closely related *Bacillus* species in clusters D and E, with *B. cereus* ATCC14579 as an outgroup [46], revealed four clades, with the species of interest distributed as follows:Clade I: *B. subtilis*, *B. spizizenii*.Clade II: *B. amyloliquefaciens*, *B. siamensis*.Clade III: *B. licheniformis*.

Thus, it can be proposed that the PhyD-Bs-1 isoform of phytase originated in a common ancestor of *B. siamensis*, *B. subtilis*, *B. spizizenii*, and *B. cereus*, and subsequently evolved, potentially influenced by specific ecological niches or other factors leading to the formation of new isoform variants that are species-specific, such as those found in *B. amyloliquefaciens*, *B. licheniformis*, and *B. cereus*.

### 3.2. Identification of PhyD Class Phytase-Encoding Genes

For the amplification of genes encoding PhyD-class phytases, we used genomic DNA from the *B. subtilis* UQM 41285, *B. cereus* ATCC 11778, *B. licheniformis* var. *mycoides* 537, and *B. amyloliquefaciens* (B10986, VKPM) strains, and primer pairs listed in Appendix A (Materials and Methods). Gel electrophoresis was used to identify DNA fragments corresponding to approximately 1150 base pairs. The nucleotide sequences of the genes encoding alkaline phytases (PhyD) were sequenced using the Sanger method and compared with nucleotide sequences available in the GenBank database (URL: http://www.ncbi.nlm.nih.gov, accessed on 1 March 2023) via BLAST. The resulting amino acid sequences, based on their mutual homology, were identified as four phytase isoforms, two of which exhibit a high degree of identity (92%) to each other. The two isoforms were derived through speciation events involving synonymous or missense mutations in *B. cereus* and *B. subtilis* (Isoform 1) and *B. amyloliquefaciens*, *B. licheniformis*, and *B. cereus* (Isoform 2). The remaining two isoforms are unique to *B. amyloliquefaciens* and *B. licheniformis* (Table 1). The identity percentage of the obtained amino acid sequences of the phytase isoforms ranged from 65% to 71% (Figure 2).

### 3.3. Generation of Integrative Genetic Constructs Containing the Phytase Gene Using the VDAC Promoter

To create a recombinant *Y. lipolytica PO1f* strain producing intracellular phytase of the PhyD class, genetic constructs containing genes encoding the phytase isoforms PhyD-Bs-1 and PhyD-Bs-2 were designed (Appendix A). The genetic system used as a basis for these constructs was described by Epova et al. [29], which allows for the generation of recombinant *Y. lipolytica* strains suitable for cultivation on low-cost industrial ingredients. Specifically, the integrative vector pUV-LT3 was employed, which contains the mitochondrial porin *VDAC* promoter of *Y. lipolytica* and the terminator of the alkaline protease gene *XPR*. This vector can be used both for intracellular expression and for the secretion of target proteins in *Y. lipolytica*.

The integrative vector was modified by inserting the PhyD-Bs-1 phytase gene, which lacks 26 amino acids from the N-terminus corresponding to the secretory leader sequence. This modification resulted in the loss of the product’s ability to be secreted outside the cell, leading to its accumulation in the cytoplasm (Figure 3). To enhance the cytoplasmic stability of the product in yeast, an additional GGA codon, which encodes glycine (Gly), was added to the N-terminus of the modified phytase gene, immediately following the ATG (Met) initiation codon, by the Warshavsky rule. The absence of substitutions in the target sequence of the modified tPhyD-Bs-1 gene and the correctness of the recombinant construct assembly were confirmed by Sanger sequencing. The resulting construct was named pUV3-tPhyD-Bs-1.

### 3.4. Selection of Y. lipolytica PO1f Transformants Carrying the Integrated Construct Expressing Phytase

The selection of transformants was performed on a minimal synthetic medium free of uracil, with the addition of leucine, at +28 °C, in two stages. The colonies that grew (~20 clones) were further organized by plating on a YNB minimal medium enriched with 1% glucose and leucine (Figure 4A).

The presence of the desired insertion (~1100 bp) was confirmed using PCR with primers specific to the modified PhyD gene (Pr-Phy-Bs/168/siamensis-f_BamHI and Pr-Phy-Bs/168/siamensis_NotI) and the high-fidelity thermostable polymerase Q5. Chromosomal DNA isolated from 20 clones (Figure 4A) was used as the template, and a yeast kit was employed for rapid isolation of chromosomal DNA suitable for PCR fragment production of at least 10,000 bp from small amounts of yeast biomass.

As shown in Figure 4B, chromosomal integration of the fragment containing the PhyD gene, under the *VDAC* promoter, occurred in most transformants, with four transformants lacking the desired fragment. To obtain “pure” transformants with the new genotype, six individual clones were transferred to fresh plates with minimal YNB medium enriched with 1% glucose and leucine (Figure 4A). PCR with the primers Pr-Vdac-Forw (targeting the promoter) and M13 fwd (after the terminator) and with primers specific for the PhyD gene confirmed the chromosomal integration of the fragment containing URA3, VDACpr-tPhyD-Bs-1, and *TXPR*. A fragment of approximately 1800 bp (URA3) and ~1100 bp (PhyD) was visualized in all the selected transformants.

Thus, the obtained transformants *Y. lipolytica PO1f* (pUV3-TPhyD-Bs-1) contained the phytase genes, as confirmed by genetic analysis (Figure 5(B.1,B.2)) and total protein analysis by electrophoresis (Figure 5C). Evaluation of phytase activity in the homogenates of the transformants’ biomass revealed that these transformants exhibited no phytase activity, likely due to intracellular protein misfolding and the formation of insoluble inclusion bodies.

### 3.5. Refolding of Phytase from Inclusion Bodies of Y. lipolytica Transformants

The refolding of the protein was performed according to a previously developed method for recovering active phytase [39]. The studies showed that proline (but not arginine) was able to recover active phytase from the washed inclusion bodies, with activity being restored immediately upon the proline addition. Changing the time and temperature conditions for denaturation in 8M urea revealed that a 30 min exposure at room temperature increased the phytase activity compared to milder conditions (20 min on ice). However, incubating the protein for several days at +10 °C resulted in a loss of activity.

The aggregation and refolding were conducted on samples from two *Y. lipolytica PO1f* (pUV3-TPhyD-Bs-1) transformants: *Y. lipolytica PO1f* (pUV3-TPhyD-Bs-1)_5 and *Y. lipolytica PO1f* (pUV3-TPhyD-Bs-1)_6. The results are shown in Table 2 and Table 3.

Denaturation of the samples with urea on ice for 20 min resulted in the appearance of phytase activity after dilution of the sample and a brief incubation with proline for 40 min (Table 2 and Table 3). As shown, the highest phytase activity was observed in both transformants after 30 min of urea treatment at room temperature, followed by 40 min of incubation with proline after dilution of the sample to a protein concentration of 0.1 mg/mL (Table 2 and Table 3). Subsequent incubation with proline at 10 °C for 3–4 days significantly reduced the initial enzyme activity in the transformants *Y. lipolytica PO1f* (pUV3—TPhyD-Bs-1)_5 and *Y. lipolytica PO1f* (pUV3—TPhyD-Bs-1)_6, respectively.

The expression level of the gene-encoding phytase was determined in the samples that were subsequently used for phytase refolding. The results of the analyses are shown in Appendix A. In both transformants, we observed a high level of expression of the PhyD-Bs-1 gene, but in transformant 5, the gene expression was 2-fold higher than in transformant 6.

We have used a new method, in which protein aggregation is monitored by measuring turbidity at 340 nm (Table 4). The turbidity (or optical density) of a solution is proportional to the size and number of protein aggregates in solution (optical density = absorbance + light scattering) resulting from light scattering in UV–visible spectroscopic measurements. Turbidity is measured within the 320–400 nm wavelength range because proteins generally had no significant absorption in the wavelength range and the magnitude of the light scattering signal increases as the wavelength decreases. Due to its simplicity, the turbidity determination method can be easily performed in a high throughput multi-well plate format. In accelerated stability testing, the propensity of proteins to aggregate in different formulations can be assessed using the kinetic method (measuring turbidity changes as a function of time at a constant temperature) [47].

When urea was added to protein extracts from transformants, which were then used for refolding, we observed a decrease in turbidity compared to that in the control (Polf) indicating a decrease in the amount of aggregate particles in solution.

## 4. Discussion

### 4.1. Advantages of PhyD-Class Phytases and the Use of Encapsulated Enzyme Forms

Microbial PhyD-class phytases, particularly those of *Bacillus* origin, demonstrate a pH optimum of 7.0–7.8 and high thermal stability, making them strong candidates for application in compound feed production. Additionally, BPPhys exhibit superior thermal stability, proteolytic resistance, and absolute substrate specificity, positioning them as an ideal replacement for commercially available phytases [7]. Phytases with a β-propeller structure have been mainly isolated from species of the *Bacillus* genus and are non-glycosylated proteins with a high thermal stability, optimal pH in the neutral range, and optimal temperature within 55 °C and 70 °C [48,49,50,51]. The phytases from *Bacillus* are suitable enzymes for the diet of monogastric animals as feed additives since their optimal pH is close to neutral. In addition, the phytases are stable at high temperatures upon granulating the feeds [50].

Before, Tran et al. (2009) reported the cloning of thermostable alkaline PhyC from the isolated *B. subtilis* MD2 into *E. coli* [25]. In 2010, Guerrero-Olazarán et al. studied the cloning and expression of the *Bacillus C* phytase gene in *P. pastoris* and showed that both recombinant and native phytases depended on calcium concentration and pH [52]. In 2007, the cloned phytase was used in animal feed. The authors demonstrated that they improved their phosphorus nutrition and reduced phosphorus contamination of animal feces [53]. The *E. coli* pET expression system could express phytases from *E. coli* and *B. subtilis* [54].

The expressed phytase of *B. subtilis* reached 20% of the total amount of soluble proteins in *E. coli* [55]. Cloning and expression of the native gene encoding the *B. subtilis* phytase using the methylotrophic yeast *Pichia pastoris* as a host organism were presented in the study [52]. The effect of N-glycosylation on biochemical features of the *B. subtilis* phytase, the impact of pH on the thermal stability of recombinant and native phytases, and the resistance of both phytases to digestive enzymes were described. The recombinant strain produced and secreted 0.82 U/mL (71 mg/L) of recombinant phytase. The enzyme was N-glycosylated, and had a molecular weight of 39 kDa. It showed an activity within the pH from 2.5 to 9.0 and at the temperatures from 25 to 70 °C, with a high residual activity of 85% after 10 min of heat treatment at 80 °C and pH of 5.5 in the presence of 5 mM CaCl_2_. The enzyme was resistant to shrimp digestive enzymes and pork trypsin. In a recent paper [18], a PCR-amplified DNA fragment being 1171 base pairs long corresponding to the PhyC gene from *B. subtilis* was also cloned upon the transformation of the *E. coli* host. The clones were confirmed using colony PCR and restriction cleavage of positive clones using XbaI/XhoI restrictase to finally confirm subcloning.

In the present study, we attempted to create a *Y. lipolytica*-based producer strain capable of synthesizing intracellular PhyD-class phytase derived from *Bacillus* species. This approach combines all the advantages of encapsulated enzymes with a pH activity optimum of 6.0–7.2, perfectly aligning with the parameters observed in the duodenal and intestinal chyme of animals. Using in silico analysis of annotated *Bacillus* genomes from the NCBI database (e.g., *B. siamensis* GCA_000262045.1, *B. subtilis* GCA_000009045.1, *B. spizizenii* GCA_000227465.1, *B. amyloliquefaciens* GCA_019396925.1, *B. cereus* GCA_002220285.1, and *B. licheniformis* GCA_034478925.1), we identified five major isoforms of beta-propeller phytase (PhyD)-encoding genes. The use of a pair of Pr-Phy-Bs/168/siamensis-f_BamHI (TTGGATCCATGGGACATTATGTGAATGAG) and Pr-Phy-Bs/168/siamensis_NotI (GGGCGGCCGCCTAGCCGTCAGAACGGTCT) primers allowed to cut off a fragment encoding a region of 26 codons long corresponding to the secretory leader. As a result, phytase loses the ability to secrete into the environment and accumulates in the cytoplasm. The primer of Pr-Phy-Bs/168/siamensis-f_BamHI provided the introduction of the BamHI restriction site, and the introduction of an artificial GGA (Gly) codon at the N-terminus of the modified tPhyD-Bs-1 gene, which, being just after the initiator codon ATG according to the Varshavsky rule, increases the cytoplasmic stability of the product in the yeast. Interestingly, one isoform, PhyD-Bs-2, was experimentally detected in four of the studied strains (*B. subtilis*, *B. licheniformis*, *B. amyloliquefaciens*, and *B. cereus*), suggesting it may have originated from a common evolutionary ancestor of these strains. In contrast, the other phytase isoforms likely evolved in response to specific environmental conditions encountered by individual *Bacillus* species or other external factors. Structural analysis of the phytase proteins revealed that all isoforms belong to BPPs. These proteins share a common architecture, forming a six-blade propeller structure (Figure 6), and including an N-terminal signal sequence of 27–30 amino acids. Notably, unlike other classes, PhyD phytases exhibit a significantly high affinity for calcium ions (Ca^2+^). The study by Ha et al. [48] provided insights into the crystal structure of a thermostable phytase complexed with calcium ions and demonstrated the critical role of these ions in enhancing the enzyme’s thermal stability and functional activity. The first blade is marked in yellow, and the second blade in green. The calcium ions Ca1, Ca2, and Ca3 are indicated in red, while the calcium ions Ca4, Ca5, and Ca6 are represented by blue dots. The blades 1 through 6 are highlighted in green. The calcium ions Ca1, Ca2, and Ca3 (marked by red dots in Figure 6) are three key calcium ions that play a critical role in ensuring the high thermal stability of the protein, occupying regions with high calcium affinity. The N-terminal segment forms an additional β-strand that connects with the fifth blade (see Figure 6). This arrangement, referred to as the “double clasp,” tightly compresses the ring-like structure, likely enhancing the enzyme’s stability. At the center of this “double clasp,” near the sixth blade, are two calcium ions (Ca1 and Ca2), which form a bivalent calcium center, where the carboxylate group of Asp 308 serves as a bridge. Ion Ca1 is coordinated with three carboxylate groups (Glu 43, Asp 308, and Asp 341), as well as with Asn 339, the carbonyl oxygen of Ile 340, and one water molecule. Ion Ca2, in turn, is coordinated with Asp 308, and Asn 336, two carbonyl oxygen atoms (from Gly 309 and Glu 338), and two water molecules. These calcium ions help reinforce the “double clasp” and contribute to the stability of the ring-like structure. Additionally, a third calcium ion (Ca3) is present in the central channel, where it coordinates with Asp 56, the carbonyl oxygen atoms of Pro 57 and Val 101, as well as three water molecules.

Additionally, Ca5 is connected to Glu 260, which is located near Ca6. Ca6, in turn, coordinates with Asp 258, Glu 260, and Gln 279 (Figure 7). The authors emphasize that the binding of ions Ca4, Ca5, and Ca6 leads to a transition from an unordered conformation of side chains to a more organized one, while the main chain structure remains unchanged. At the top of the phytase structure, a shallow cleft is formed, predominantly filled with negatively charged residues. Binding of Ca^2+^ to specific amino acids (low-affinity sites) likely enhances phytase stability, which, in turn, may improve its enzymatic activity.

In the paper in reference [56], the biochemical features of bacterial histidine acid phytase from *Pantoea* sp. 3.5.1 obtained in three different expression systems including the *Y. lipolytica* yeast were compared. The highest activity of the AgpP recombinant phytase expressed by *Y. lipolytica* was observed at an acidic pH (pH of 3.0). The recombinant enzyme of AgpP-Y remained active within pH from 3.0 to 7.0, inactivating at a pH of more than 8.0. However, the temperature optimum of the AgpP-Y recombinant phytase was of 45 °C. Divalent metal ions at a concentration of 1 mM affected similarly the recombinant phytase activity, namely Ca^2+^, Mg^2+^, and Mn^2+^ ions doubled the activity of the enzyme. The data indicate a high potential for the application of the *Y. lipolytica* yeast as an object for the expression of heterologous phytases of bacterial origin.

Previously, we proposed a unique technology for producing PhyC-class phytase derived from the enterobacterium *O. proteus*. The key advantages of this enzyme include its near-neutral pH optimum, which ensures high efficiency in the animal intestine, and its high thermal stability when encapsulated directly within the cells of the recombinant producer *Y. lipolytica* [4]. The production of encapsulated phytase is also waste-free, as both the biomass of the producer and the residual protein from the medium are not discharged as waste but are instead utilized as animal feed. This makes the production process highly cost-effective, even with relatively low enzyme yields. Furthermore, an additional advantage of the feed enzyme encapsulated in *Y. lipolytica* is its protection against degradation by stomach acid and pepsin. The solubilization of *Y. lipolytica* cells and the release of the enzyme occur only in the duodenum.

We hope this unique technology, which was developed before for the only object of the phytase from *O. proteus,* can be also designed as a feed supplement for the animals using the phytase from bacteria of the *Bacillus* genus.

### 4.2. Effect of Structural Features of PhyC and PhyD Phytases on Their Folding Process in Y. lipolytica

Phytases from different classes, like other proteins, possess unique physicochemical properties that can vary depending on their amino acid sequence and three-dimensional structure. The proper organization and spatial arrangement of amino acid residues determine enzyme activity, their ability to bind substrates and metal ions, and their stability under various physicochemical conditions. Structural differences between the PhyC and PhyD phytases, such as the number of cysteines (disulfide bonds), the presence of glycosylation sites, and the hydrophobicity index of the amino acid sequence, play an important role in the protein folding process, especially in heterologous systems. These factors can influence both the thermodynamic stability and the functionality and catalytic activity of the enzymes. One key aspect of this process is the positioning of hydrophobic and hydrophilic residues, which affect intermolecular interactions and the formation of intermediate states during protein folding. A comparison of the structural characteristics of PhyC and PhyD phytases (Table 5) revealed significant differences in the number of cysteines, glycosylation sites, and the quantity of hydrophobic and hydrophilic residues, which may explain the differences in their folding capabilities.

Thus, in a paper referenced at [57], the X-ray structure of phytase from *Aspergillus ficuum* was published. The authors confirmed that there are five disulfide bonds in the protein, namely Cys8-S17, Cys48-Cys391, Cys192-Cys442, Cys241-Cys259, and Cys413-Cys421. Ula and Mullaney [58] reported that the denatured GuHCl phytase could restore its activity when the denaturant was diluted in a reactivation system without beta-mercaptoethanol at a pH of 7.5 but could not restore its activity when added a reducing agent. The changes in intrinsic fluorescence spectra, CD spectra, and enzyme activity were measured to study the role of disulfide bonds when the denaturation occurred in urea. It was found that disulfide bonds play an essential role in the three-dimensional structure and catalytic activity of the enzyme [59]. The results showed that disulfide bonds are necessary for the structure and activity in the phytase of *Aspergillus* sp.

Although disulfide bonds in phytase are considered important to maintain the conformation and activity upon unfolding, only a few attempts have been made to study the role of disulfide bonds in conformational change upon reactivation and refolding. In particular, there is no direct evidence of either a conformational change or a link between structural changes and biological features of the enzyme. In the study at [60], amino acid residues of BPP from *B. subtilis* 168 (168PhyA), Ser-161, and Leu-212 were mutated into cysteine residues using site-directed mutagenesis and the effects of the engineered disulfide bond in *B. subtilis* 168 PhyA were studied. Although the double cysteine mutant was secreted from *B. subtilis* at an expression level, which was 3.5 times higher than that in the wild type, neither biochemical nor enzymatic features of the enzymes changed. In enzymatic assays, the mutant phytase showed poor refolding ability after thermal denaturation. The authors stated that the disulfide bond in the BPP sequences of gram-negative bacteria is beneficial for their stability in the periplasmic compartment. On the contrary, the lack of periplasmic space in *Bacillus* species and the fact that *Bacillus* BPP are released extracellularly can make disulfide bonds unnecessary. This may explain why, during the evolution BPP in *Bacillus* species carry no disulfide bonds.

Protein aggregation, along with the structural features of β-propeller type phytases is the main obstacle encountered in the production of recombinant proteins. Unproductive protein aggregations can appear both due to non-specific (hydrophobic) interactions of predominantly unfolded polypeptide chains and due to improper interactions between partially structured folding intermediates. Hydrophobic interactions may be the main reason for protein aggregation, as there is no possibility for disulfide bonds due to the absence of cysteine residues in phytase [39].

Protein aggregation can be suppressed by various co-solvents used, for example, proline. Artificial chaperones such as glycerol and proline applied for refolding increase the viscosity of the solution, which can modulate the dynamics of protein refolding so that partially folded intermediates could successfully complete the folding process rather than being blocked by aggregation [39].

Osmolytes such as proline can mainly affect intermediate protein products with a partially disrupted network of internal hydrophobic interactions that decreases the nonpolar surface exposed to the solvent. Osmolytes are known to initiate the restoration of the internal hydrophobic core of a protein for proper folding. Kumar et al. [61] suggested that proline behaves as a chaperone of protein folding due to the formation of an amphipathic supramolecular assembly. The production of *Bacillus* phytases in active form in *E. coli* also was a limited success as the enzyme was sequestered in the inclusion bodies [49]. In 2008, in the work of Rao et al. [39], a new PhyC phytase was isolated and cloned from *B. subtilis* in *E. coli*. The authors attempted to isolate the active enzyme from the inclusions and describe the recombinant PhyC. However, the type of PhyC was produced at a slower rate [7].

In this study, we applied the technology of intracellular heterologous expression of the PhyD class phytase from *Bacillus*, which had previously been successfully used to produce PhyC class phytase from *Enterobacteriaceae O. proteus* in the cytoplasm of recombinant *Y. lipolytica* yeast cells. However, unlike PhyC (Phy-OP), which is synthesized in its active form, PhyD class phytase (PhyD-Bs-1, PhyD-Bs-2) tends to aggregate during intracellular accumulation. The aggregation of PhyD class phytase may depend on several key factors, including high expression levels, a lack of cysteines, and oxidative stress (Table 5). PhyC phytase from *O. proteus* has a unique structure consisting of eight cysteine residues, which significantly contribute to its stability and functionality. In contrast to Phy-OP, only one isoform of PhyD class phytases, PhyD-Bs-3, contains a single cysteine, while the other isoforms are completely devoid of cysteine (Table 5). During protein folding, the proper alignment of the polypeptide chain largely depends on intermolecular interactions between various amino acids, including the formation of disulfide bonds between cysteine residues. Since cysteines are lacking in PhyD class phytases, this can hinder protein stabilization through such bonds, thereby increasing the likelihood of misfolded conformations and aggregates. These processes are especially likely under oxidative stress conditions when proteins are more prone to denaturation and aggregation. The absence of cysteines (Table 5) may also increase the time required for folding, as proteins are forced to find less efficient pathways to form stable structures without disulfide bonds. This can cause a mismatch between the rate of synthesis and the need for folding, which may result in an increased likelihood of producing misfolded protein forms. Therefore, using a strong promoter to express PhyD phytase in *Y. lipolytica* may significantly increase the chances of producing incompletely folded protein (misfolding). Oxidative stress, in turn, contributes to the formation of free radicals (reactive oxygen species), and if the protein molecule lacks SH-groups, it loses its protective mechanism usually provided by antioxidants and other oxidative reactions. Under these conditions, there is a greater need for chaperones—molecules that assist proteins in adopting the correct conformations. However, when there is an excess of misfolded proteins, chaperones may become overloaded, leading to further accumulation of aggregates.

### 4.3. Protein Aggregation and Refolding of PhyD Phytase for Biotechnological Applications

Non-productive protein aggregation can arise both from the non-specific hydrophobic interactions characteristic of unfolded polypeptide chains and from incorrect interactions between partially structured folding intermediates. To address this issue, protein refolding using artificial chaperones, such as glycerol and proline, helps partially folded intermediates complete the folding process without blocking aggregation. Osmolytes, such as proline, affect intermediate protein products with partially disrupted internal hydrophobic interactions, reducing the exposed non-polar surface area available for solvent interaction. This helps restore the hydrophobic core of the protein and ensures correct folding.

We successfully performed the refolding of PhyD phytase synthesized in *Y. lipolytica* cells using osmolytes like proline, which overcame the aggregation problem and yielded a functionally active product. In our studies, protein refolding was performed using proline, which could restore active phytase from the washed inclusion bodies and the activity was restored just after the amino acid addition (Table 2 and Table 3). The results confirmed the assumption of chaperone function of proline upon the reactivation of phytase obtained from the inclusion bodies in our experiments.

During the research, we developed an effective protocol for cloning, expression, and in vitro refolding of the Bacillus phytase enzyme from eukaryotic organisms. The resulting enzyme, which exhibits a broad pH range, high thermostability, the ability to renature through refolding procedures, and substrate specificity, appears promising as an efficient feed additive. In pilot-scale experiments, the transformed *Y. lipolytica PO1f* (pUV3—TPhyD-Bs-1) strain, expressing the phytase gene during bioreactor cultivation, shows potential as a suitable candidate for large-scale production of active phytase enzyme.

## 5. Conclusions

In this study, we studied the intracellular heterologous expression of PhyD phytase from *Bacillus* species in *Y. lipolytica* yeast cells. While this technology has been successfully used to synthesize PhyC phytase from *O. proteus*, PhyD phytase tends to aggregate during intracellular accumulation. The primary factors contributing to PhyD phytase aggregation are high protein expression levels, the absence of cysteines, and oxidative stress conditions. Unlike PhyC phytase, which contains eight cysteine residues ensuring stability, only one PhyD isoform (PhyD-Bs-3) contains cysteine, while the others lack this amino acid entirely. This absence hinders the formation of disulfide bonds, making PhyD phytase more prone to misfolding and aggregation. These issues are particularly critical under oxidative stress, which increases the likelihood of protein denaturation.

Additionally, the use of a strong promoter for PhyD phytase expression can create an imbalance between protein synthesis and folding, further increasing the risk of producing misfolded forms. Without protective mechanisms, such as SH groups, proteins become more vulnerable to damage from free radicals. Chaperones, which help in proper folding, may become overloaded, worsening the aggregation problem. One approach to improving PhyD phytase folding is the incorporation of glycosylation, which can significantly enhance its stability and activity. Specifically, adding N-glycosidic bonds helps prevent protein degradation and improves functional characteristics.

Through refolding PhyD phytase, synthesized in *Y. lipolytica* cells, using osmolytes (e.g., proline), we successfully overcame aggregation issues and obtained a functionally active product. This opens up new possibilities for applying PhyD phytases in various biotechnological fields. Protein aggregation, such as that of PhyD phytase, remains a significant challenge in the production of recombinant proteins in heterologous systems. However, strategies to optimize codon usage (such as the gene transfer of PhyD from prokaryotes like *Bacillus* to eukaryotes like *Y. lipolytica*), create disulfide bonds, and incorporate glycosylation—combined with the use of osmolytes—can significantly improve protein stability and efficiency. These approaches make PhyD phytases more promising for biotechnological applications, opening new possibilities for their use in both scientific and industrial settings.

## Figures and Tables

**Figure 1 jof-11-00186-f001:**
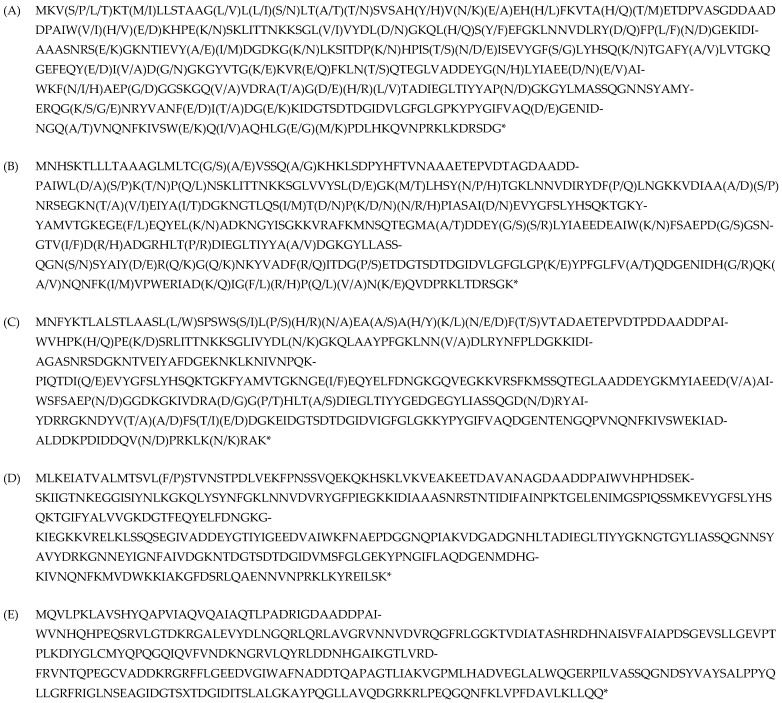
Phytase isoforms from *Bacillus* species are represented in the NCBI database (https://www.ncbi.nlm.nih.gov/, accessed on 11 March 2023). The amino acid variability at each position is indicated in parentheses. *—stop codon. (**A**) Length: 383 aa. Species: *B. siamensis*, *B. subtilis*, *B. spizizenii*, *B. cereus*. Amino acid variability: ~16%, identity: ~84%. (**B**) Length: 384 aa. Species: *Bacillus amyloliquefaciens*, *B. siamensis*. Amino acid variability: ~13%, identity: ~87%. (**C**) Length: 382 aa. Species: *B. licheniformis*. Amino acid variability: ~7%, identity: ~93%. (**D**) Length: 391 aa. Species: *B. cereus*. Amino acid variability: ~0.3%, identity: ~99.7%. (**E**) Length: 335 aa. Species: *B. subtilis*. Identity: 100%.

**Figure 2 jof-11-00186-f002:**
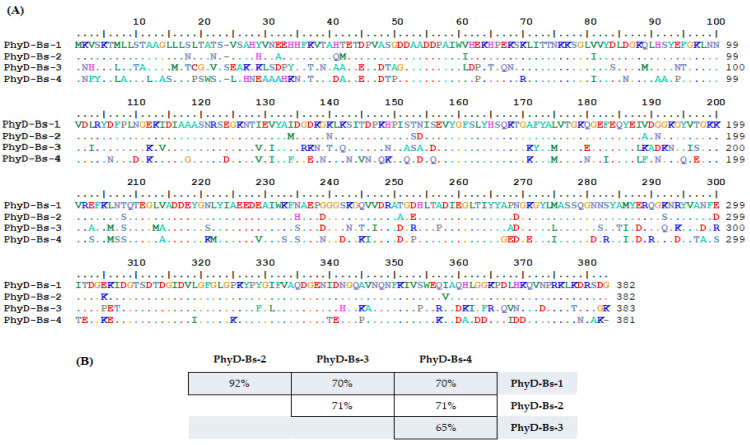
Comparison of phytase isoform amino acid sequences. (**A**) Amino acid sequences obtained by translating the nucleotide sequences for four isolated phytase isoforms. A 100% identity is marked by dots; (**B**) percent identity between amino acid sequences of isolated phytase isoforms.

**Figure 3 jof-11-00186-f003:**
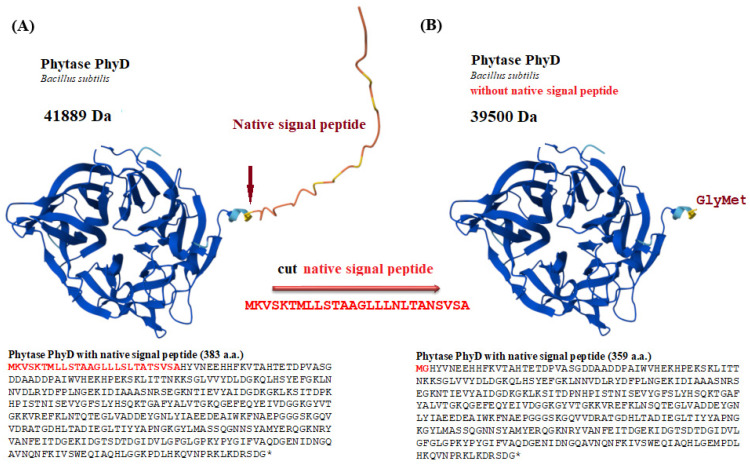
Amino acid sequence and structure of phytase PhyD (AlphaFold Number AF-P42094-F1-v4, UniProt Number P42094). With native signal peptide (**A**); without native signal peptide (**B**). *—stop codon. The structure of beta-propeller phytase (BPPs) was predicted using Alpha fold software (https://alphafold.ebi.ac.uk/download, accessed on 1 June 2023).

**Figure 4 jof-11-00186-f004:**
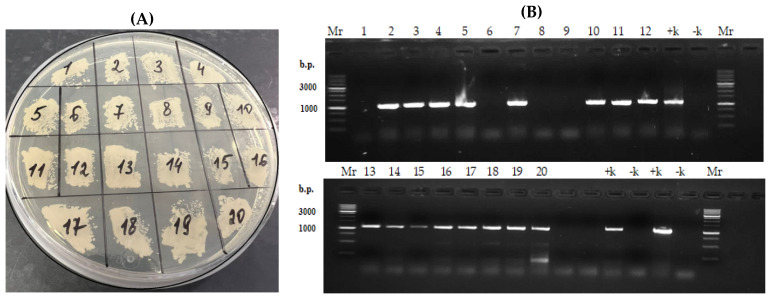
Selection of *Y. lipolytica PO1f* transformants carrying the integrated construct pUV3-tPhyD-Bs-1. (**A**) Primary selection on YNB minimal medium enriched with 1% glucose and leucine, free of uracil. (**B**) Electrophoresis in a 1.2% agarose gel. Lanes: Mr—DNA marker 1 kb (Eurogen NL001), 1–20—PCR products (~1100 bp) from chromosomal DNA isolated from transformants, +k—PCR from pUV3-tPhyD-Bs-1 vector, −k—PCR from PO1f strain.

**Figure 5 jof-11-00186-f005:**
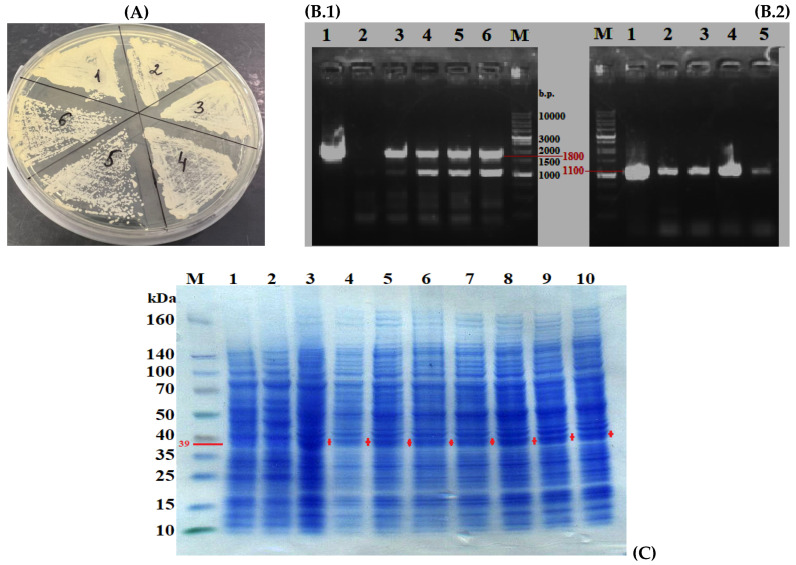
Secondary selection of transformants with the new genotype. (**A**) Individual colonies of transformants were replated on minimal YNB medium, enriched with 1% glucose and leucine, and uracil-free. (**B.1**) Electrophoresis in 1.2% agarose gel. Amplification results using primers Pr-Vdac-Forw and Pr-M13 fwd (~1800 bp): 1—vector pUV3-tPhyD-Bs-1, 2—chromosomal DNA from strain Polf, 3–6—chromosomal DNA from transformants, M—DNA marker 1 kB (Eurogen NL001). (**B.2**) Electrophoresis in a 1.2% agarose gel. Amplification results using primers Pr-Phy-Bs/168/siamensis-f_BamHI and Pr-Phy-Bs/168/siamensis_NotI (~1100 bp): 1—vector pUV3-tPhyD-Bs-1, 2—chromosomal DNA from strain Polf, 3–5—chromosomal DNA from transformants, M—DNA marker 1 kB (Eurogen NL001). (**C**) Protein electrophoresis in a polyacrylamide gel: 1—strain POLf, 2—strain W29, 3–10—transformants with the new genotype producing phytase (~39,500 Da). The red pluses indicate the location of the target protein.

**Figure 6 jof-11-00186-f006:**
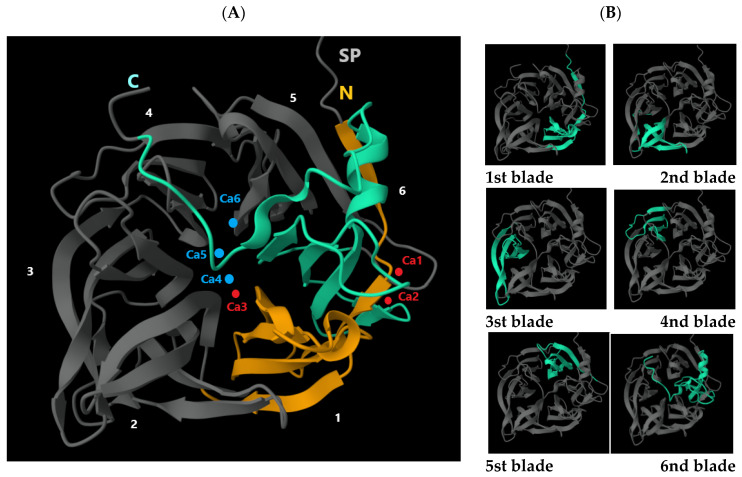
Structure of BPPs. UniProt Number Q2I2M8. AlphaFold Number AF-Q2I2M8-F1-v4. (**A**)—N-terminal signal sequence of 27–30 amino acids. The first blade is marked in yellow, and the second blade in green. The calcium ions Ca1, Ca2, and Ca3 are indicated in red, while the calcium ions Ca4, Ca5, and Ca6 are represented by blue dots. (**B**)—green marked each blade from the six-bladed propeller structure.

**Figure 7 jof-11-00186-f007:**
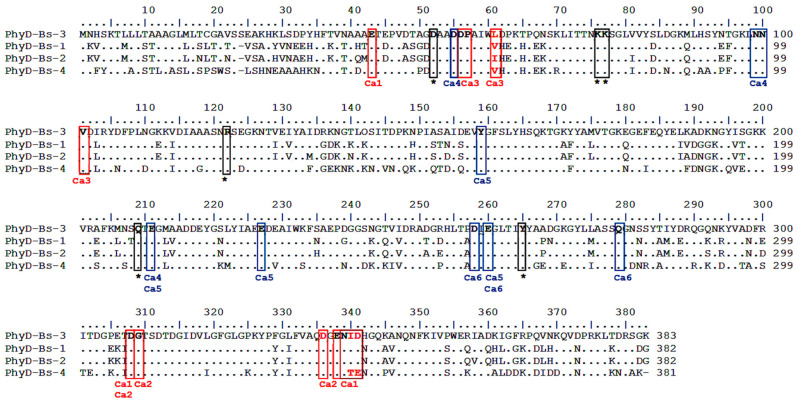
Comparison of amino acid residues interacting with calcium ions in four phytase isoforms. Amino acid residues interacting with ions Ca1, Ca2, and Ca3 are highlighted in red, while residues interacting with ions Ca4, Ca5, and Ca6 are highlighted in blue. (*) indicates indirect effects.

**Table 1 jof-11-00186-t001:** Phytase isoforms identified in four *Bacillus* Species.

Isolated Isoform	Organism Scientific Name	Sequence (a.a) of Phytase (PhyD), *—Stop Codon	Length (a.a.)	Identical Proteins, Number in GenBank	Percent Identity, %	Isoform In Silico
PhyD-Bs-1	▪ *B. subtilis UQM 41285* ▪ *B. cereus* *ATCC* *11778*	MKVSKTMLLSTAAGLLLSLTATSVSAHYVNEEHHFKVTAHTETDPVASGDDAADDPAIWVHEKHPEKSKLITTNKKSGLVVYDLDGKQLHSYEFGKLNNVDLRYDFPLNGEKIDIAAASNRSEGKNTIEVYAIDGDKGKLKSITDPKHPISTNISEVYGFSLYHSQKTGAFYALVTGKQGEFEQYEIVDGGKGYVTGKKVREFKLNTQTEGLVADDEYGNLYIAEEDEAIWKFNAEPGGGSKGQVVDRATGDHLTADIEGLTIYYAPNGKGYLMASSQGNNSYAMYERQGKNRYVANFEITDGEKIDGTSDTDGIDVLGFGLGPKYPYGIFVAQDGENIDNGQAVNQNFKIVSWEQIAQHLGGKPDLHKQVNPRKLKDRSDG*	383	▪ *B. subtilis* WP_326253589.1 ▪ *Alkalicoccobacillus gibsonii* MBU8842768.1 ▪ *B. tequilensis* AQX83395.1	100% 100% 98%	PhyD-isoform-1
PhyD-Bs-2	▪ *B. subtilis UQM 41285* ▪ *B. cereus ATCC 11778* ▪ *B. amyloliquefaciens (B10986, BKПM)* ▪ *B. licheniformis var. mycoides 537*	MKVSKTMLLSTAAGLLLNLTANSVSAHHVNAEHHFKVTAQMETDPVASGDDAADDPAIWIHEKHPEKSKLITTNKKSGLIVYDLDGKQLHSYEFGKLNNVDLRYDFPLNGEKIDIAAASNRSEGKNTIEVYAMDGDKGNLKSITDPKHPISSDISEVYGFSLYHSQKTGAFYALVTGKQGEFEQYEIADNGKGYVTGKKVREFKLNSQTEGLVADDEYGNLYIAEEDEAIWKFHAEPDGGSKGQVVDRAAGEHLTADIEGLTIYYAPDGKGYLMASSQGNNSYAMYERQGSNRYVANFDITDGKKIDGTSDTDGIDVLGFGLGPKYPYGIFVAQDGENIDNGQAVNQNFKIVSWEQVAQHLGGKPDLHKQVNPRKLKDRSDG*	383	▪ *B. spizizenii WP_003218192.1* ▪ *B. rugosus WP_166851630.1* ▪ *B. subtilis WP_326253589.1* ▪ *B. inaquosorum WP_268289454.1* ▪ *B. vallismortis* *WP_268535344.1*	100% 95% 94% 93% 93%	PhyD-isoform-1
PhyD-Bs-3	▪ *B. amyloliquefaciens (B10986, VKPM)*	MNHSKTLLLTAAAGLMLTCGAVSSEAKHKLSDPYHFTVNAAAETEPVDTAGDAADDPAIWLDPKTPQNSKLITTNKKSGLVVYSLDGKMLHSYNTGKLNNVDIRYDFPLNGKKVDIAAASNRSEGKNTVEIYAIDRKNGTLQSITDPKNPIASAIDEVYGFSLYHSQKTGKYYAMVTGKEGEFEQYELKADKNGYISGKKVRAFKMNSQTEGMAADDEYGSLYIAEEDEAIWKFSAEPDGGSNGTVIDRADGRHLTPDIEGLTIYYAADGKGYLLASSQGNSSYTIYDRQGQNKYVADFRITDGPETDGTSDTDGIDVLGFGLGPKYPFGLFVAQDGENIDHGQKANQNFKIVPWERIADKIGFRPQVNKQVDPRKLTDRSGK*	383	▪ *B. siamensis* WP_045926145.1 ▪ *B. amyloliquefaciens* WP_269389756.1 *B. velezensis* WP_201488970	100% 96% 96%	PhyD-isoform-2
PhyD-Bs-4	▪ *B. licheniformis var. mycoides 537*	MNFYKTLALSTLAASLLSPSWSSLSHNEAAAHKNFTVTADAETEPVDTPDDAADDPAIWVHPKHPEKSRLITTNKKSGLIVYDLNGKQLAAYPFGKLNNVDLRYNFPLDGKKIDIAGASNRSDGKNTVEIYAFDGEKNKLKNIVNPQKPIQTDIQEVYGFSLYHSQKTGKFYAMVTGKNGEIEQYELFDNGKGQVEGKKVRSFKMSSQTEGLAADDEYGKMYIAEEDVAIWSFSAEPNGGDKGKIVDRADGPHLTADIEGLTIYYGEDGEGYLIASSQGDNRYAIYDRRGKNDYVTAFSTEDGKEIDGTSDTDGIDVIGFGLGKKYPYGIFVAQDGENTENGQPVNQNFKIVSWEKIADALDDKPDIDDQVNPRKLKNRAK*	382	▪ *B. haynesii WP_154995585.1* ▪ *B. licheniformis* *AFQ59979.1* ▪ *B. paralicheniformis WP_079289540.1*	100% 96% 97%	PhyD-isoform-3
Non-identified	▪ *B. cereus ATCC 11778*	MLKEIATVALMTSVLFSTVNSTPDLVEKFPNSSVQEKQKHSKLVKVEAKEETDAVANAGDAADDPAIWVHPHDSEKSKIIGTNKEGGISIYNLKGKQLYSYNFGKLNNVDVRYGFPIEGKKIDIAAASNRSTNTIDIFAINPKTGELENIMGSPIQSSMKEVYGFSLYHSQKTGIFYALVVGKDGTFEQYELFDNGKGKIEGKKVRELKLSSQSEGIVADDEYGTIYIGEEDVAIWKFNAEPDGGNQPIAKVDGADGNHLTADIEGLTIYYGKNGTGYLIASSQGNNSYAVYDRKGNNEYIGNFAIVDGKNTDGTSDTDGIDVMSFGLGEKYPNGIFLAQDGENMDHGKIVNQNFKMVDWKKIAKGFDSRLQAENNVNPRKLKYREILSK*	390	▪ *B. cereus* *PEY43863.1* ▪ *B. pseudomycoides WP_040119248.1* ▪ *B. arachidis* *WP_081904783.1*	100% 95% 93%	PhyD-isoform-4
Non-identified	▪ *B. cereus ATCC 11778*	MQVLPKLAVSHYQAPVIAQVQAIAQTLPADRIGDAADDPAIWVNHQHPEQSRVLGTDKRGALEVYDLNGQRLQRLAVGRVNNVDVRQGFRLGGKTVDIATASHRDHNAISVFAIAPDSGEVSLLGEVPTPLKDIYGLCMYQPQGQIQVFVNDKNGRVLQYRLDDNHGAIKGTLVRDFRVNTQPEGCVADDKRGRFFLGEEDVGIWAFNADDTQAPAGTLIAKVGPMLHADVEGLALWQGERPILVASSQGNDSYVAYSALPPYQLLGRFRIGLNSEAGIDGTSXTDGIDITSLALGKAYPQGLLAVQDGRKRLPEQGQNFKLVPFDAVLKLLQQ*	334	▪ *B. cereus* *AHM26864.1* ▪ *Gallaecimonas pentaromativorans* *WP_170164176.1* *WP_050658825.1*	100% 97% 98%	PhyD-isoform-5

**Table 2 jof-11-00186-t002:** Phytase activity in transformants *Y. lipolytica PO1f* (pUV3-TPhyD-Bs-1)_5 after refolding.

Sample	Isolation Stage and Incubation Conditions	Phytase Activity, U/mg of Protein
1	After urea treatment on ice for 20 min, followed by incubation for 40 min with proline	0.5 ± 0.01 *
2	After urea treatment for 30 min at room temperature and incubation with proline for 40 min.	0.94 ± 0.04 *
3	3 days of incubation with proline.	0.05 ± 0.001
4	4 days of incubation with proline at +10 °C.	0

Means of * are significantly different, *p* ≤ 0.05.

**Table 3 jof-11-00186-t003:** Phytase activity in transformants *Y. lipolytica PO1f* (pUV3—TPhyD-Bs-1)_6 after refolding.

Sample	Isolation Stage and Incubation Conditions	Phytase Activity, U/mg of Protein
1	Urea treatment on ice for 20 min followed by incubation with proline for 40 min	0.26 ± 0.01 *
2	After urea treatment for 30 min at room temperature and incubation with proline for 40 min.	0.47 ± 0.01 *
3	3 days of incubation with proline.	0.06 ± 0.001
4	4 days of incubation with proline at +10 °C.	0

Means of * are significantly different, *p* ≤ 0.05.

**Table 4 jof-11-00186-t004:** Results of turbidity determination in transformant lysates.

Samples	PO1f (MatA, leu2-270, ura3-302, xpr2-322, asp-2)	PO1f Transformant Biomass (pUV3-tPhyD-Bs-1)_5	PO1f Transformant Biomass (pUV3-tPhyD-Bs-1)_6
Turbidity OD350 urea free	0.12 ± 0.02	0.35 ± 0.07	0.24 ± 0.05
Turbidity OD350 8 M urea	0.10± 0.03	0.12 ± 0.03	0.11 ± 0.04

**Table 5 jof-11-00186-t005:** Structural characteristics of PhyC and PhyD phytases.

Phytase	Predicted N-Glycosylation Sites *	Predicted O-Glycosylation Sites **	GRAVY ^#^	Number of Cys ^$^	Total Different Combinations of S
PhyD-Bs-1	2	2	−0.597	no	0
PhyD-Bs-2	1	1	−0.618	no	0
PhyD-Bs-3	2	0	−0.620	no	1
PhyD-Bs-4	2	1	−0.699	no	0
phy-OP	0	4	−0.364	8	764

Notes: * https://services.healthtech.dtu.dk/services/NetNGlyc-1.0/, accessed on 11 March 2023, ** https://services.healthtech.dtu.dk/services/NetOGlyc-4.0/, accessed on 11 March 2023, **^#^** https://web.expasy.org/protparam/, accessed on 11 March 2023. The average hydropathy (GRAVY) of a linear polypeptide sequence is calculated by summing the hydropathy values of all amino acids and dividing by the number of residues in the sequence. An increase in the positive score indicates greater hydrophobicity, but it does not account for how the protein folds in three dimensions, or the percentage of residues buried in the protein’s hydrophobic core. ^$^
https://npsa-prabi.ibcp.fr/cgi-bin/npsa_automat.pl?page=/NPSA/npsa_cysteines.html, accessed on 11 March 2023.

## Data Availability

The original contributions presented in this study are included in the article/Appendix A. Further inquiries can be directed to the corresponding author.

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
