# Peer review of "Assessment of Recombinant β-Propeller Phytase of the Bacillus Species Expressed Intracellularly in Yarrowia lipolityca"

_jof, 2025, doi:10.3390/jof11030186_

Round 1
Reviewer 1 Report
This article identified several PhyD phytases from Bacillus species and intracellularly expressed one of them in Yarrowia lipolityca using the integrative vector pUV-LT3. To recover active phytase from washed inclusion bodies, the authors treated the protein with urea and proline, successfully overcoming aggregation issues and obtaining a functionally active product. Thus, I would recommend accepting and publishing it after some revisions.
Detailed comments:
[1] Page 3, lines 103-106: “In the proposed technology there is no stage of separation biomass from the cultural medium…”, the detailed proposed technology to obtain the protein intracellularly expressed in Y. lipolityca should be mentioned. In addition, the active phytase produced in this research was recovered from purified inclusions. Biomass separation from the cultural medium and cell lysis were required in the production of active phytase. As a result, “no stage of separation biomass from the cultural medium” should not be an advantage of intracellular phytase production. Please revise it.
[2] Page 7: “As a result of bioinformatics (comparative) analysis, we identified five variants of phytases,…”, what is the protein used as for the query sequence to identify phytases? In addition, are the five variants of phytases and their isoforms (PhyD-Bs-1, 2, 3, and 4) available in the database? Please provide the NCBI or GenBank accession numbers of these proteins.
[3] There are mistakes in the figure numbers inserted in the Results part. Page 8, lines 337-340: “Figure 1” should be replaced by “Figure 2”. Page 14, line 436 to Page 15, line 456: “Figure 6” should be replaced by “Figure 7”. Please revise them.
[4] The letter writing format of A and B in Figure 5 is inconsistent with the other pictures. The position and format of C in Figure 7 are inconsistent. Please revise them.
[5] The data provided in Tables 3 and 4 lack variance and significance analyses. Therefore, it is impossible to analyze the reproducibility of the results and the effectiveness of the refolding optimization. Please supply the necessary data.
[6] The phyD-Bs-2 wasn’t characterized in the research. Why constructed the phyD-Bs-2 expression plasmid in section 3.3?
[7] The sources of the protein structures in Figures 5 and 8 are not provided. Please provide the PDB codes or the structural simulation processes.
[8] Is phyD-Bs-1 a novel protein that had not been characterized by previous studies? Does phyD-Bs-1 have a pH optimum of 7.0–7.8 and high thermal stability which were mentioned in the Introduction and Discussion parts? Please provide more activity and stability data.
This article identified several PhyD phytases from Bacillus species and intracellularly expressed one of them in Yarrowia lipolityca using the integrative vector pUV-LT3. To recover active phytase from washed inclusion bodies, the authors treated the protein with urea and proline, successfully overcoming aggregation issues and obtaining a functionally active product. Thus, I would recommend accepting and publishing it after some revisions.
Detailed comments:
[1] Page 3, lines 103-106: “In the proposed technology there is no stage of separation biomass from the cultural medium…”, the detailed proposed technology to obtain the protein intracellularly expressed in Y. lipolityca should be mentioned. In addition, the active phytase produced in this research was recovered from purified inclusions. Biomass separation from the cultural medium and cell lysis were required in the production of active phytase. As a result, “no stage of separation biomass from the cultural medium” should not be an advantage of intracellular phytase production. Please revise it.
[2] Page 7: “As a result of bioinformatics (comparative) analysis, we identified five variants of phytases,…”, what is the protein used as for the query sequence to identify phytases? In addition, are the five variants of phytases and their isoforms (PhyD-Bs-1, 2, 3, and 4) available in the database? Please provide the NCBI or GenBank accession numbers of these proteins.
[3] There are mistakes in the figure numbers inserted in the Results part. Page 8, lines 337-340: “Figure 1” should be replaced by “Figure 2”. Page 14, line 436 to Page 15, line 456: “Figure 6” should be replaced by “Figure 7”. Please revise them.
[4] The letter writing format of A and B in Figure 5 is inconsistent with the other pictures. The position and format of C in Figure 7 are inconsistent. Please revise them.
[5] The data provided in Tables 3 and 4 lack variance and significance analyses. Therefore, it is impossible to analyze the reproducibility of the results and the effectiveness of the refolding optimization. Please supply the necessary data.
[6] The phyD-Bs-2 wasn’t characterized in the research. Why constructed the phyD-Bs-2 expression plasmid in section 3.3?
[7] The sources of the protein structures in Figures 5 and 8 are not provided. Please provide the PDB codes or the structural simulation processes.
[8] Is phyD-Bs-1 a novel protein that had not been characterized by previous studies? Does phyD-Bs-1 have a pH optimum of 7.0–7.8 and high thermal stability which were mentioned in the Introduction and Discussion parts? Please provide more activity and stability data.
Author Response
Answers to the REVIEWER 1
This article identified several PhyD phytases from Bacillus species and intracellularly expressed one of them in Yarrowia lipolityca using the integrative vector pUV-LT3. To recover active phytase from washed inclusion bodies, the authors treated the protein with urea and proline, successfully overcoming aggregation issues and obtaining a functionally active product. Thus, I would recommend accepting and publishing it after some revisions.
We thank the reviewer for the high evaluation of our study
Detailed comments:
[1] Page 3, lines 103-106: “In the proposed technology there is no stage of separation biomass from the cultural medium…”, the detailed proposed technology to obtain the protein intracellularly expressed in Y. lipolityca should be mentioned. In addition, the active phytase produced in this research was recovered from purified inclusions. Biomass separation from the cultural medium and cell lysis were required in the production of active phytase. As a result, “no stage of separation biomass from the cultural medium” should not be an advantage of intracellular phytase production. Please revise it.
Answer: We have corrected
[2] Page 7: “As a result of bioinformatics (comparative) analysis, we identified five variants of phytases,…”, what is the protein used as for the query sequence to identify phytases? In addition, are the five variants of phytases and their isoforms (PhyD-Bs-1, 2, 3, and 4) available in the database? Please provide the NCBI or GenBank accession numbers of these proteins.
Answer: The initial step involved searching for nucleotide and amino acid sequences of phytases in annotated Bacillus strain genomes available in the NCBI GenBank database (https://www.ncbi.nlm.nih.gov/). Reference sequences from studies [26] and [27], along with keyword searches (“Phytase.. Bacillus”, “beta-propeller phytase Bacillus”), were used. Orthologs of PhyD-class phytase genes were identified using NCBI NucleotideBLAST and ProteinBLAST tools (https://blast.ncbi.nlm.nih.gov/Blast.cgi). All retrieved sequences (Figures S1–S5) were analyzed via multiple sequence alignment using Clustal W [28] in BioEdit v6.0.7, followed by classification into isoforms exhibiting high amino acid residue homology (Figures S6–S9). Universal primers were designed for each isoform (Table 1) to amplify phytase genes from the following Bacillus strains: B. subtilis UQM 41285 (ATCC 23857, strain 168), B. cereus ATCC 11778, B. licheniformis var. mycoides 537, and B. amyloliquefaciens (B10986, VKPM). Sequence alignments were generated with Clustal W [28] in BioEdit v6.0.7. Clustal W parameters included sequence weighting, position-specific gap penalties, and weight matrix selection to enhance alignment sensitivity [28].
To understand better the experimental results and to compare them with the results of bioinformatic analysis (in silico), we supplemented and reformatted the table as follows:
The isolated isoform is the name of phytase isoforms amplified with genomic DNA from the Bacillus strains.
Organizm Scientific name is the name of the Bacillus strains, which were the sources forphytaseisoforms
Sequence (a.a) of phytase (PhyD) is full amino acid sequences using the translation from nucleotide sequences using BioEdit software version 6.0.7 [Thompson JD, Higgins DG, Gibson TJ: CLUSTAL W: improving the sensitivity of progressive multiple sequence alignment through sequence weighting, position-specific gap penalties and weight matrix choice. Nucleic Acids Res. 1994, 22 (22): 4673-4680. 10.1093/nar/22.22.4673.]
Length (a.a.) is length of amino acid sequences of the phytase
Identical proteins Number in GenBank is search for identical amino acid sequences in the NCBI database using the NCBI – NucleotideBLAST (https://blast.ncbi.nlm.nih.gov/Blast.cgi ). The NCBI access number is shown.
Percent identity, % is the percent of the identity of the amino acid sequence of the phytaseisoformsobtained as aresult of an experimentwith those in the GenBankNCBIdatabase.
Isoform In silico is the column of the data shows which version of the isoform (Figure 2) due to bioinformatic analysis is of the maximum homology.
[3] There are mistakes in the figure numbers inserted in the Results part. Page 8, lines 337-340: “Figure 1” should be replaced by “Figure 2”. Page 14, line 436 to Page 15, line 456: “Figure 6” should be replaced by “Figure 7”. Please revise them.
Answer: We have corrected.
[4] The letter writing format of A and B in Figure 5 is inconsistent with the other pictures. The position and format of C in Figure 7 are inconsistent. Please revise them.
Answer: We have corrected.
[5] The data provided in Tables 3 and 4 lack variance and significance analyses. Therefore, it is impossible to analyze the reproducibility of the results and the effectiveness of the refolding optimization. Please supply the necessary data.
Answer: We have done.
[6] The phyD-Bs-2 wasn’t characterized in the research. Why constructed the phyD-Bs-2 expression plasmid in section 3.3?
Answer: A comparison of the amino acid sequences of the PhyD-Bs-1 and PhyD-Bs-2 isoforms obtained in this work showed that the isoforms have high similarity – about 94% of the identity, as a result, constructs for both isoforms were obtained. In the study, we focused on one of the isoforms of PhyD-Bs-1.
[7] The sources of the protein structures in Figures 5 and 8 are not provided. Please provide the PDB codes or the structural simulation processes.
Answer: In our work, we used protein structures from the AlphaFold Protein Structure Database as a model, which have an amino acid sequence identity with PhyD-Bs-1 of more than 96%.
AlphaFold Number AF-Q2I2M8-F1-v4, UniProt Number Q2I2M8 (with signal sequence), AlphaFold Number AF-P42094-F1-v4, UniProt Number P42094 (with signal sequence), AlphaFold Number AF-Q9F657-F1-v4, UniProt Number Q9F657 (without signal sequence), In the article, we indicated the numbers in AlphaFold and UniProt.
PDB codes - None available in the PDB
Structures from the program were saved in tiff format.
Figure 5. Amino Acid Sequence and Structure of Phytase PhyD-Bs-1A (AlphaFold Number AF-P42094-F1-v4, UniProt Number P42094). With Native Signal Peptide B. Without Native Signal Peptide. (AlphaFold Number AF-Q9F657-F1-v4, UniProt Number Q9F657)
Figure 8. Structure of beta-propeller phytase (BPPs). UniProt Number Q2I2M8. AlphaFold Number AF-Q2I2M8-F1-v4
[8] Is phyD-Bs-1 a novel protein that had not been characterized by previous studies? Does phyD-Bs-1 have a pH optimum of 7.0–7.8 and high thermal stability which were mentioned in the Introduction and Discussion parts? Please provide more activity and stability data.
Answer: We have not tested the features in the study. Our task was to show a crutial possibility for getting the incapsulated bacterial phytase based on the Yarrowia lipolytica yeast and designing of the technology for obtaining of the active enzyme after its refolding.
Reviewer 2 Report
The manuscript goal is to clone the coding sequence of phytases from different Bacillus species and produce them in the cytoplasm of Yarrowia lipolytica for vet nutritional use of the whole cell containing the enzyme. That is an interesting and relevant research however not totally original. The authors present the importance of their work and foccus on one of the enzyme forms.
However, the Material and Methods description is quite confuse. Some information is given repeatidly, other are missing and there are some mistakes. Also, there is a need for more references of what is described in the section (and in other sections too).
Results section describe data and show figures and tables which methodology was not presented in Methods section, as in silico analysis for example.
Discussion section is based in only a few references and do not show the real importance of the reference result with the result of the present work. In Discussion and Conclusion sections, the authors insisted on affirm that low enzyme activity is based on protein aggregation due to high expression and absence of disulphide bonds. However, they do not show the existence of aggregates and do not confirm the high expression level of the recombinant DNA.
Language improvement could also contribute to the better understanding of the work.
The work is interesting and relevant but needs to go through a major revision to be acceptable for publishing in a Journal with quality like JoF.
Following, there are some detailed data that needs to be revised for publication:
- Please look through all the manuscript because there are a lot of places where scientific names, such as Bacillus, are not in italic formatting (examples: line 62, 114, 120 etc);
- in line 78 it is stated that PhyC has lower thermal stability but the opposite is in line 73;
- line 113: there is a comma before Moreover;
- line 135: Is the mentioned kit correct? Did you use a kit for RNA extraction from blood cells to extract the genomic DNA of a gram positive bacteria?
- line 136: in Fig. 1 it is written 1.2% agarose but in line 136 it is 1%. Which is the correct agarose concentration?
- line 140, Fig 1 title: electrophoresis of what? Please describe what is shown in the gel;
- line 145: this topic 2.3 is not necessary since Table 1 is mentioned again in topic 2.4. Please leave only topic 2.4;
- Table 1: please provide one more column with Genbank access number of each gene;
- line156: you do not need to repeat here that primer pairs are shown in Table 1 because it is written above;
- line 157: change matrix for template;
- line 158: this topic (2.5. PCR Conditions) could be embedded in 2.4 topic;
- line 166: since the enzymes will not have the secretory signal, they are truncated forms. It is recommended to use that terminology and standard the writting of gene description as tphyD-Bs-1 throughout the text;
- line 167: change matrix for template;
- lines 171 to 179: all this part is a discussion. Please remove it from Methodology;
- line 176: please give the reference to the Varshavsky rule;
- line 185: change mQ for ultrapure water;
- lines 188 and 189: use only mcg or the micro symbol, not both for units;
- line 196: since it is an integrative vector, which enzyme did the authors use to linearize the vector for chromosomal integration?
- line 196: why two recombinant DNA were constructed (lines 193 and 194) but only one was transformed to the yeast?
- line 197: genotype and provenience of the yeast was given before (lines 131-132). Please do not repeat the information;
- line 219: use TXPR with XPR in italic form;
- line 219-220: please, avoid giving results in the Methodology section ("In all... 1800 bp long" is result);
- line 268: correct pH6/0;
- line 280 to 386: it is important to give the method used in the in silico search and alignment shown as first result, which is absent is Methodology section;
- Figure 2: to the better understanding of this result, it should be presented a global alignment and a phylogenetic tree, as given by some softwares as Mega. Online tools such as Clustal omega could also be used and identity could be added in a table;
- lines 335 to 360: the result stated in those lines is impossible to see by the result presented as it is now. Please, change the result Figure 2 for a global alignment of the enzyme sequences and add a phylogenetic analysis (by using Mega software, for example). The phylogenetic tree is a good form to show the evolutionary relationship among the enzyme sequences;
- Table 2 and Figure 3: Table 2 is not since the sequences should be deposited in Genbank. The accession number of the deposits are relevant results. Enzyme sequences and size should be visualized in the alignment, as in Figure 3A. Please remove formatting marks that are now present at the end of each line of the alignment. Table shown in Fig 3B should be a table, not a figure, and should be self explanatory. Besides, title and legend and Figure 3 should be better described;
- Figure 4: In the vector map, please indicate promoter and cds with an arrow, as suggested by standard features of the software used to produce the figure (Snapgene). Add "VDACpr" and "TXPR" to indicate where the promoter and terminator are, respectivelly. Indicate the software name used to produce the vector map;
- lines 401-410: information in lines 169-179 should complement the discussion here;
- line 404: the obtaining of figure 5 should be described. Was the modeling based on a crystal structure? Which one? Give the PDB code access. Which software was used? Or did you use Alpha fold?
- line 407: Please give a reference for this rule. What is the difference between the Warsaw rule and the Varshavsky rule cited in line 176?
- line 408: again, the enzyme sequence was truncated, therefore, it should be named showing that information and the representation should indicate the truncation, like tphyD-Bs. Also, please standard how to write the sequence name. Sometimes it is BS, others are Bs. Since it is a bacterial gene, phy should alwaysbe used in italics and lower case while the enzyme only could be Phy, with P upper case;
- lines 429, 452, 453: correct figure number;
- line 430 and 431: The kit reported in Methodology was named as a kit for RNA extraction from blood cells. Please give correct information about the kit used;
- line 439: better named as VDACpr and TXPR (with gene names in italic);
- lines 440 to 451: please, indicate figure 7;
- lines 453 to 456: recombinant protein electrophoresis of yeast inocula is not informative since protein prodduction difference is too subtle. Please change figure 7C to a western blot for the construction containing phy or to a pre-purification of the enzyme to make it more reliable of the protein production;
- table 3 and 4: what does it mean "_5" and "_6" in the strain identification? Are those two different colonies tested for refolding and activity? If so, since the activity of 6 is half of 5, more colonies should be tested. Also, was it used the same protein concentration of both samples to refold and test activity?
- lines 508 to 511: these Genbank accession number should be presented previously, as already mentioned in this Comment section;
- lines 514 and 515: as suggested, phylogenetic analysis will be important to confirm such affirmation;
- line 522: which is the homology % of the phytase cloned in this work and the phytase described in reference 37?
- Figure 8: Information about how the figure was obtained should be provided, such as software used and modeling based on which PDB crystal;
- lines 529 to 531: I supose this is Fig 8 legend. Please indicate it properly;
- line 602: correct table number;
- Table 5: this in silico analysis should be described in methodology;
- lines 620 to 633: this part is very similar to what is written in 4.2 section. Please write just once the corresponding information;
- line 640: please, show the protein aggregation;
- Discussion in general: there is a lack of literature to give basis in discussion and to the hypothesis of the absence of activity in the cloned enzyme. Only a few papers are mentioned with no clear correlation. For example, why references 32 and 7 are cited when enzyme glycosilation is mentioned? Other example, a big part of topic 4.1 is all written based only in reference 37 but it is not clear why it is necessary to explore too much about the calcium positioning in the enzyme since not a single methodology was used to analyse the metal position or influence in the activity in the present study.
- lines 674 to 682: reference for the information given here is needed too;
- line 691: I could not find where the authors say in Methodotology that a bioreactor was used. Please make the enzyme production clear;
- line 698: nor aggregation neither expression level of the recombinant enzyme were demonstrated in the present work. After refolding, activity is still low. Also, if the protein is endogenous of a bacteria, what does it make active in bacteria but is lacking in yeast? Without not a single expression level analysis, there is no evidence of misfolding due to high expression;
Author Response
Answers to REVIEWER 2
Description of Methods section is confused and have some mistakes. Some methods are too short and could be embedded in other method. Some results are shown with Figures but the respective method is not described
Answer: We have revised the methods according to the recommendations and additionally included a method for protein expression studies (RNA isolation, cDNA synthesis, primer design, PCR, qPCR) and a method for protein aggregation studies based on the change in turbidity of protein extract in the presence of urea (turbidimetry).
Some results could be presented in different figures or tables to help understanding and avoid mistakes in the analysis by anyone. Some results are reported in Results or Discussion section but the method used to reach that analysis is not shown. There is a great lack of references in Discussion. Some topics in the Discussion are presented in two or more big paragraphs with only one reference cited. Authors Conclusion explore protein aggregation due to high expression level of the recombinant DNA but there are no confirmation neither of the aggregation nor of the high expression. All conclusion is only based in hypothesis of expression level, lack of disulphide bonds and aggregation, without any experimental data to confirm that.
Answer: We have revised the Results and Discussion sections as it was recommended and have included the data on additional experiments to investigate the recombinant phytase expression and changes in protein extract turbidity in the presence of 8M urea.
The contribution of the article is based in the cloning of a phytase D. Indeed, it is not clear if the authors clone and express a phytase C and a D or only the D form. The cloning data could be more explored, including by an evolutionary point of view, but in that sense, the publication in JoF could not be recommended. To provide a relevant contribution for publication in JoF, the expression, production and activity of the cloned phytase should be confirmed. In that sense, it is necessary to show data of: expression level of the recombinant enzyme sequence; western blot or a simple protein purification to confirm the production of the phytase in the yeast; after yeast cell lysis and separation of supernatant and pellet, there is a need to shown the presence of the enzyme in the pellet to shown its presence as aggregate; To reach the activity goal, after yeast lysis, the proteins could be at least concentrated to explore the enzyme activity with and without refolding.
Answer: We performed some additional experiments, which are reported in all the sections of the paper (methods, results, and discussion).
Some Figures make it difficult to see the results, there are table as Figure, other figure has formatting markers. Other figures should be added to the better understanding of the paper. All of that are described in Detail comments.
Answer: Additional material has been added, namely Figures S1-S10. The information about software, GenBank numbers, etc. has been added to the tables and figures.
The manuscript goal is to clone the coding sequence of phytases from different Bacillus species and produce them in the cytoplasm of Yarrowia lipolytica for vet nutritional use of the whole cell containing the enzyme. That is an interesting and relevant research however not totally original. The authors present the importance of their work and foccus on one of the enzyme forms. However, the Material and Methods description is quite confuse. Some information is given repeatidly, other are missing and there are some mistakes. Also, there is a need for more references of what is described in the section (and in other sections too).
Answer: We have corrected the methods section according to the recommendations, and added the necessary information.
Results section describe data and show figures and tables which methodology was not presented in Methods section, as in silico analysis for example.
Answer: We have added
Discussion section is based in only a few references and do not show the real importance of the reference result with the result of the present work. In Discussion and Conclusion sections, the authors insisted on affirm that low enzyme activity is based on protein aggregation due to high expression and absence of disulphide bonds. However, they do not show the existence of aggregates and do not confirm the high expression level of the recombinant DNA.
Answer: After lysis of yeast cells and separation of the supernatant and the precipitate we showed the presence of the enzyme in the precipitate. We studied the expression level of the recombinant enzyme sequence in the yeast to demonstrate its presence as an aggregate. To achieve the activity after yeast lysis proteins could be concentrated to study the enzyme activity with and without refolding. We have revised the Discussion section, including the necessary references and shortening some sections.
Language improvement could also contribute to the better understanding of the work.
Answer: We have revised using an native-speaking scientist
The work is interesting and relevant but needs to go through a major revision to be acceptable for publishing in a Journal with quality like JoF.
Answer: We are grateful to the reviewer for the remarks and recommendations on improving the manuscript
Detail comments
Following, there are some detailed data that needs to be revised for publication:
- Please look through all the manuscript because there are a lot of places where scientific names, such as Bacillus, are not in italic formatting (examples: line 62, 114, 120 etc);
Answer: We have corrected.
- in line 78 it is stated that PhyC has lower thermal stability but the opposite is in line 73;
Answer: We have corrected the statement in line 78
- line 113: there is a comma before Moreover;
Answer: We have replaced the comma for the dot
- line 135: Is the mentioned kit correct? Did you use a kit for RNA extraction from blood cells to extract the genomic DNA of a gram positive bacteria?
Answer: We have corrected the typo. High quality total genomic DNA was isolated using the ExtractDNA Blood & Cells kit (BC111M, Total DNA extraction kit for whole blood, animal cells and bacteria, Evrogen, Russia) according to the manufacturer's protocol.
- line 136: in Fig. 1 it is written 1.2% agarose but in line 136 it is 1%. Which is the correct agarose concentration?
Answer: We have corrected the concentration for 1.2 %
- line 140, Fig 1 title: electrophoresis of what? Please describe what is shown in the gel;
Answer: We have modified Figure 1 and added information to the description
- line 145: this topic 2.3 is not necessary since Table 1 is mentioned again in topic 2.4. Please leave only topic 2.4;
Answer: We have corrected
- Table 1: please provide one more column with Genbank access number of each gene;
Answer: We have added the GenBank/UniProtKB accession number to Table 1 for each primer and we have added the Supplementary Figures S6-S10, where the GenBank numbers can also be seen.
- line156: you do not need to repeat here that primer pairs are shown in Table 1 because it is written above;
Answer: We have corrected
- line 157: change matrix for template;
Answer: We have done
- line 158: this topic (2.5. PCR Conditions) could be embedded in 2.4 topic;
Answer: We have united topics 2.4 and 2.5
- line 166: since the enzymes will not have the secretory signal, they are truncated forms. - It is recommended to use that terminology and standard the writting of gene description as tphyD-Bs-1 throughout the text;
Answer: We have corrected the name of isoforms without secretory signal to tphyD-Bs-1, tphyD-Bs-2.
- line 167: change matrix for template;
Answer: Construction of an integrative expression vector
- lines 171 to 179: all this part is a discussion. Please remove it from Methodology;
Answer: We have transferred the abstract into the Discussion section
- line 176: please give the reference to the Varshavsky rule;
Answer: Alexander Varshavsky “The N-end rule pathway and regulation by proteolysis”, PROTEIN SCIENCE 2011, VOL 20:1298—1345
- line 185: change mQ for ultrapure water;
Answer: We have done
- lines 188 and 189: use only mcg or the micro symbol, not both for units;
Answer: We have done
- line 196: since it is an integrative vector, which enzyme did the authors use to linearize the vector for chromosomal integration?
Answer: For chromosomal integration, the vector was linearized with BglII enzyme (R0144M, NEB), and the restriction site is shown in the vector map (Fig. 4).
- line 196: why two recombinant DNA were constructed (lines 193 and 194) but only one was transformed to the yeast?
Answer: Comparison of the amino acid sequences of PhyD-Bs-1 and PhyD-Bs-2 isoforms obtained in this study showed that the isoforms have high similarity, about 94% identity. In this study, we focused on one of the PhyD-Bs-1 isoforms.
- line 197: genotype and provenience of the yeast was given before (lines 131-132). Please do not repeat the information;
Answer: We have done
- line 219: use TXPR with XPR in italic form;
Answer: We have done
- line 219-220: please, avoid giving results in the Methodology section ("In all... 1800 bp long" is result);
Answer: We have deleted: In all the selected transformants there was a fragment 1800 bp long
- line 268: correct pH6.0;
Answer: We have done, pH 6.0
- line 280 to 386: it is important to give the method used in the in silico search and alignment shown as first result, which is absent is Methodology section;
Answer: P. 2.2. In silico bioinformatics analysis
The initial step involved searching for nucleotide and amino acid sequences of phytases in annotated Bacillus strain genomes available in the NCBI GenBank database (https://www.ncbi.nlm.nih.gov/). Reference sequences from studies EF536824 [32] and U85968 [15], along with keyword searches (“Phytase.. Bacillus”, “beta-propeller phytase Bacillus”), were used. Orthologs of PhyD-class phytase genes were identified using NCBI NucleotideBLAST and ProteinBLAST tools (https://blast.ncbi.nlm.nih.gov/Blast.cgi).
All retrieved sequences (Figures S1–S5) were analyzed via multiple sequence alignment using Clustal W [49] in BioEdit v6.0.7, followed by classification into isoforms exhibiting high amino acid residue homology (Figures S6–S9). Universal primers were designed for each isoform (Table 1) to amplify phytase genes from the following Bacillus strains: B. subtilis UQM 41285 (ATCC 23857, strain 168), B. cereus ATCC 11778, B. licheniformis var. mycoides 537, and B. amyloliquefaciens (B10986, VKPM).
Sequence alignments were generated with Clustal W [49] in BioEdit v6.0.7. Clustal W parameters included sequence weighting, position-specific gap penalties, and weight matrix selection to enhance alignment sensitivity [Thompson JD, et al. Nucleic Acids Res. 1994;22(22):4673-4680. doi:10.1093/nar/22.22.4673].
- Figure 2: to the better understanding of this result, it should be presented a global alignment and a phylogenetic tree, as given by some softwares as Mega. Online tools such as Clustal omega could also be used and identity could be added in a table;
Answer: In the description, we referred to articles, where a global alignment of phytases and Bacillus strains was already performed and various phylogenetic trees were designed. We relied on the published phylogenetic trees and high similarity of phytase isoforms obtained in our work with the phytases, for which phylogenetic analysis has already been performed.
In the study, we believe that the information provided is quite sufficient, since it is clear, which clusters the isoforms obtained will belong to.
- lines 335 to 360: the result stated in those lines is impossible to see by the result presented as it is now. Please, change the result Figure 2 for a global alignment of the enzyme sequences and add a phylogenetic analysis (by using Mega software, for example). The phylogenetic tree is a good form to show the evolutionary relationship among the enzyme sequences;
Answer: Bioinformatic analysis of the amino acid sequences of PhyD phytases from Bacillus strains represented in the NCBI GenBank database is presented more clearly in the appendix (Figures S1-S5).
- Table 2 and Figure 3: Table 2 is not since the sequences should be deposited in Genbank. The accession number of the deposits are relevant results. Enzyme sequences and size should be visualized in the alignment, as in Figure 3A. Please remove formatting marks that are now present at the end of each line of the alignment. Table shown in Fig 3B should be a table, not a figure, and should be self-explanatory. Besides, title and legend and Figure 3 should be better described;
Answer: We have corrected.
To better understand the experimental data and correlate it with the results of bioinformatics analysis (in silico), we have expanded and reformatted the table as follows:
Isolated Isoform – Name of the phytase isoforms amplified from the genomic DNA of Bacillus strains.
Organism Scientific Name – Name of the Bacillus strains from which the phytase isoforms were experimentally obtained.
Sequence (a.a.) of Phytase (PhyD) – Full amino acid sequences obtained by translating nucleotide sequences using BioEdit software version 6.0.7 [Thompson JD, Higgins DG, Gibson TJ: CLUSTAL W: improving the sensitivity of progressive multiple sequence alignment through sequence weighting, position-specific gap penalties, and weight matrix choice. Nucleic Acids Res. 1994, 22 (22): 4673-4680. doi:10.1093/nar/22.22.4673].
Length (a.a.) – Length of the phytase amino acid sequence.
Identical Proteins Number in GenBank – Search for identical amino acid sequences in the NCBI database using the NCBI NucleotideBLAST tool (https://blast.ncbi.nlm.nih.gov/Blast.cgi). The NCBI accession number is provided.
Percent Identity, % – Percentage identity of the experimentally obtained phytase isoforms compared to sequences available in the NCBI GenBank database.
Isoform In Silico – This column indicates which isoform variant (see Figure 2) identified through bioinformatics analysis shows the highest homology.
Figure 3. Comparison of Phytase Isoform Amino Acid Sequences. (A) Amino acid sequences obtained by translating the nucleotide sequences for four isolated phytase isoforms. A 100% identity is marked by dots; (B) Percent identity between amino acid sequences of the isolated phytase isoforms.
- Figure 4: In the vector map, please indicate promoter and cds with an arrow, as suggested by standard features of the software used to produce the figure (Snapgene). Add "VDACpr" and "TXPR" to indicate where the promoter and terminator are, respectivelly. Indicate the software name used to produce the vector map;
Answer: We have added
Figure 4. Map of the Integrative Vector pUV3-tPhyD-Bs for Expression of Bacillus Phytases (tPhyD-Bs-1, tPhyD-Bs-2) Under the VDAC Promoter of Y. lipolytica. (Software SnapGene Viewer (5.0.7)).
- lines 401-410: information in lines 169-179 should complement the discussion here;
Answer: We have done
- line 404: the obtaining of figure 5 should be described. Was the modeling based on a crystal structure? Which one? Give the PDB code access. Which software was used? Or did you use Alpha fold?
Answer: The structure of beta-propeller phytase (BPPs) shown in Figure 5 was modeled using Alpha fold software. The structures of phytases, which are of high similarity in amino acid sequences with the isoforms obtained in this study were taken. UniProtKB accession: Q2I2M8, AlphaFold: AF- Q2I2M8-F1-v4; UniProtKB accession: P42094 AlphaFold: AF-P42094-F1-v4; UniProtKB accession: Q9F657, AlphaFold: AF-Q9F657-F1-v4, (without signal sequence) are 98% similar in amino acid sequence to Phy-Bs-1 and 94% similar to Phy-Bs-2. UniProtKB accession: O66037, AlphaFold Number: AF-O66037-F1-v4 is similar to Phy-Bs-3 by 96 %. The phytase UniProtKB accession: Q8KTX7, AlphaFold: AF-Q8KTX7-F1 is 95% similar to Phy-Bs-4. A crystal structure was obtained for phytase UniProtKB accession: O66037 [37] in the presence and absence of calcium ions (Protein Data Bank (accession codes 1POO, 2POO, 1CVM, and 1QLG)).
We carefully investigated the location of highly conserved amino acid residues that bind calcium ions with different affinity in the crystal structure (1POO, 2POO, 1CVM, and 1QLG) and in the predicted structures (AF- Q2I2M8-F1-v4, AF-P42094-F1-v4, AF-O66037-F1-v4 AF-Q9F657-F1-v4).
From the analysis, it was observed that the calcium-binding amino acid residues are in the same spatial position in the predicted structures. In Figure 5, we took the structure of UniProtKB accession: Q2I2M8, labeled the blade loops using Alpha fold, and showed the location of calcium ions similar to how they are arranged in the crystal structure of 1POO, 2POO, 1CVM, and 1QLG.
- line 407: Please give a reference for this rule. What is the difference between the Warsaw rule and the Varshavsky rule cited in line 176?
Answer: We have added the Ref. to Varshavsky rule
- line 408: again, the enzyme sequence was truncated, therefore, it should be named showing that information and the representation should indicate the truncation, like tphyD-Bs. Also, please standard how to write the sequence name. Sometimes it is BS, others are Bs. Since it is a bacterial gene, Phy should always be used in italics and lower case while the enzyme only could be Phy, with P upper case;
Answer: We have corrected. In the text, the designation tphyD-Bs is used for the truncated isoforms. Bs and Phy
- lines 429, 452, 453: correct figure number;
Answer: We have corrected.
- line 430 and 431: The kit reported in Methodology was named as a kit for RNA extraction from blood cells. Please give correct information about the kit used;
Answer: We have corrected the typo. High quality total genomic DNA was isolated using the ExtractDNA Blood & Cells kit (BC111M, Total DNA extraction kit for whole blood, animal cells and bacteria, Evrogen, Russia) according to the manufacturer's protocol.
- line 439: better named as VDACpr and TXPR (with gene names in italic);
Answer: We have done
- lines 440 to 451: please, indicate figure 7;
Answer: We have done
- lines 453 to 456: recombinant protein electrophoresis of yeast inocula is not informative since protein production difference is too subtle. Please change figure 7C to a western blot for the construction containing phy or to a pre-purification of the enzyme to make it more reliable of the protein production
Answer: We have added a new method, in which protein aggregation is monitored by measuring turbidity at 340 nm.
- table 3 and 4: what does it mean "_5" and "_6" in the strain identification? Are those two different colonies tested for refolding and activity? If so, since the activity of 6 is half of 5, more colonies should be tested. Also, was it used the same protein concentration of both samples to refold and test activity?
Answer: We have selected two strains (5 and 6) from 150 colonies by progressive assaying the enzyme activity. Those strains seemed most promising. In the Tables we presented only the most successful versions. Certainly we used the same protein concentrations in the experiments.
- lines 508 to 511: these Genbank accession number should be presented previously, as already mentioned in this Comment section;
Answer: We have added the Genbank accession number
- lines 514 and 515: as suggested, phylogenetic analysis will be important to confirm such affirmation;
Answer: We have added the Supplementary Figures S6-S10, where the GenBank numbers can also be seen.
Figure S1. UniProtKB: P42094, GeneBank: NC_000964; UniProtKB: A0A6M3ZCL5, GeneBank:NP_389861.1, UniProtKB: A0A5D4NBZ6, UniProtKB: A0A6A8FPB9, UniProtKB: A0A7U3NS85, UniProtKB: A0A8I2B9V6, UniProtKB: A0A0D1IPL3, UniProtKB: A0A165AEM5, UniProtKB: Q2I2M8, UniProtKB: A0A5F2KH80, UniProtKB: A0A7U3NS43
Figure S2. UniProtKB: P42094; UniProtKB: A0A5B8YW95; UniProtKB: S066037; UniProtKB: A0A2K9KG41; UniProtKB: 0A8F7V592; UniProtKB: Q8VQS1; UniProtKB: A0A0G2UMU5; UniProtKB: Q8KTX7; UniProtKB: AHM26864.1; UniProtKB: AUZ26779.1; UniProtKB: CUB29737.1
Figure S3. UniProtKB: CUB29737.1, UniProtKB: AUZ26779.1, UniProtKB: AHM26864.1, UniProtKB: AMR43729.1, UniProtKB: PEY43863.1, UniProtKB: WP_098335196.1
Figure S4. UniProtKB: A0A0G2UMU5, UniProtKB: I3QPI9, UniProtKB: J7JV12, UniProtKB: Q8KTX7, UniProtKB: A0A6C1VYY4, UniProtKB: A0A7Y6P8J0, UniProtKB: Q65NG0, UniProtKB: Q6DNH6, UniProtKB: niProtKB: A0A0N9ZNE5, UniProtKB: A4UU76, UniProtKB: T5HJS2.
Figure S5. UniProtKB: A0A2K9KG41, UniProtKB: A0A4V7TP60, UniProtKB: A0A6H3BDN1, UniProtKB: A0A8F6CZV0, UniProtKB: A0A8F7N7I6, UniProtKB: A0A8F7V592, UniProtKB: A0A8F9SJG5, UniProtKB: A0A8F9XLQ9, UniProtKB: A0A8G1LKS4, UniProtKB: A7Z5Q0, UniProtKB: F4ZE79, UniProtKB: I2C6L5, UniProtKB: M9TD51, UniProtKB: Q5MCL8, UniProtKB: Q8VQS1, UniProtKB: S6FXY0, UniProtKB: Q938A7, UniProtKB: AAL25193.3.
- line 522: which is the homology % of the phytase cloned in this work and the phytase described in reference 37?
Answer: The similarity in amino acid residues was 70% to the Phy-Bs-1 isoform and 96% to Phy-Bs-3
- Figure 8: Information about how the figure was obtained should be provided, such as software used and modeling based on which PDB crystal;
Answer: The structure of beta-propeller phytase (BPPs) shown in Figure 5 was modeled using Alpha fold software. The structures of phytases that have high similarity in amino acid sequences with the isoforms obtained in this study were taken. UniProtKB accession: Q2I2M8, AlphaFold: AF- Q2I2M8-F1-v4; UniProtKB accession: P42094 AlphaFold: AF-P42094-F1-v4; UniProtKB accession: Q9F657, AlphaFold: AF-Q9F657-F1-v4, (without signal sequence) are 98% similar in amino acid sequence to Phy-Bs-1 and 94% similar to Phy-Bs-2. UniProtKB accession: O66037, AlphaFold Number: AF-O66037-F1-v4 is similar to Phy-Bs-3 by 96 %. The phytase UniProtKB accession: Q8KTX7, AlphaFold: AF-Q8KTX7-F1 is 95% similar to Phy-Bs-4. A crystal structure was obtained for phytase UniProtKB accession: O66037 [37] in the presence and absence of calcium ions (Protein Data Bank (accession codes 1POO, 2POO, 1CVM, and 1QLG)).
We carefully investigated the location of highly conserved amino acid residues that bind calcium ions with different affinity in the crystal structure (1POO, 2POO, 1CVM, and 1QLG) and in the predicted structures (AF- Q2I2M8-F1-v4, AF-P42094-F1-v4, AF-O66037-F1-v4 AF-Q9F657-F1-v4).
From the analysis, it was observed that the calcium-binding amino acid residues are in the same spatial position in the predicted structures. In Figure 5, we took the structure of UniProtKB accession: Q2I2M8, labeled the blade loops using Alpha fold, and showed the location of calcium ions similar to how they are arranged in the crystal structure of 1POO, 2POO, 1CVM, and 1QLG.
- lines 529 to 531: I suppose this is Fig 8 legend. Please indicate it properly;
Answer: We have revised
- line 602: correct table number;
Answer: We have corrected the table number to 5
- Table 5: this in silico analysis should be described in methodology;
Answer: We have done
- 2.2. Structural Characteristics of PhyC and PhyD Phytases
- Predicted N-glycosylation sites were calculated using the program http://www.cbs.dtu.dk/services/NetNGlyc/ - N-glycosylation
- Predicted O-glycosylation sites were calculated using the program http://www.cbs.dtu.dk/services/NetOGlyc/ - O-glycosylation
- GRAVY was calculated using the program https://web.expasy.org/protparam/ - MW, aa, GRAVY, pI. The average hydropathy (GRAVY) of a linear polypeptide sequence is calculated by summing the hydropathy values of all amino acids and dividing by the number of residues in the sequence. An increase in the positive score indicates greater hydrophobicity, but it does not account for how the protein folds in three dimensions or the percentage of residues buried in the protein's hydrophobic core
- Number of Cys и Total were calculated using the program different combinations https://npsa-prabi.ibcp.fr/cgi-bin/npsa_automat.pl?page=/NPSA/npsa_cysteines.html Disulfide bonds Notes
- lines 620 to 633: this part is very similar to what is written in 4.2 section. Please write just once the corresponding information;
Answer: We have corrected
- line 640: please, show the protein aggregation;
Answer: We have added a new method, where protein aggregation is monitored by measuring turbidity at 340 nm. The turbidity (or optical density) of a solution is proportional to the size and number of protein aggregates in solution (optical density = absorbance + light scattering), resulting from light scattering in UV-visible spectroscopic measurements. Turbidity is measured in the 320-400 nm wavelength range because proteins generally do not have significant absorption in this wavelength range and the magnitude of the light scattering signal increases as the wavelength decreases. Due to its simplicity, the turbidity determination method can be easily performed in a high throughput multi-well plate format. In accelerated stability testing, the propensity of proteins to aggregate in different formulations can be assessed using either the kinetic method (measuring turbidity changes as a function of time at constant temperature [High-Throughput Biophysical Analysis of Protein Therapeutics to Examine Interrelations between Aggregate Formation and Conformational Stability Rajoshi Chaudhuri, Yuan Cheng,. Russell Middo,1 and David B. Wolkin, The AAPS Journal, Vol. 16, 2014].
Protein aggregation was monitored by measuring turbidity at 340 nm. Aggregates purified from DNA and some yeast protein impurities that were paralleled for refolding were dissolved in 5 ml of 50 mM Tris-HCl buffer, pH 7.0, analysing the amount of protein according to Bradford. Purified inclusion cells were solubilised with 8 M urea and placed 200 µl per well of a 96-well plate (Corning). Changes in turbidity were recorded at 340 nm using a Synergy HTX tablet reader (Biotek). Three biological (n > 3) and three technical (n > 3) repeats were performed for each experiment.
- Discussion in general: there is a lack of literature to give basis in discussion and to the hypothesis of the absence of activity in the cloned enzyme. Only a few papers are mentioned with no clear correlation. For example, why references 32 and 7 are cited when enzyme glycosilation is mentioned? Other example, a big part of topic 4.1 is all written based only in reference 37 but it is not clear why it is necessary to explore too much about the calcium positioning in the enzyme since not a single methodology was used to analyse the metal position or influence in the activity in the present study.
Answer: We have revised
- lines 674 to 682: reference for the information given here is needed too;
Answer: We have added
- line 691: I could not find where the authors say in Methodotology that a bioreactor was used. Please make the enzyme production clear;
Answer: We have raised the yeast in the flasks, in the laboratory conditions
- line 698: nor aggregation neither expression level of the recombinant enzyme were demonstrated in the present work. After refolding, activity is still low. Also, if the protein is endogenous of a bacteria, what does it make active in bacteria but is lacking in yeast? Without not a single expression level analysis, there is no evidence of misfolding due to high expression;
Answer: We have revised
2.10 RNA isolation and reverse transcription reaction
Total RNA from Y. lipolytica biomass was extracted using the RNeasy Mini Kit (Qiagen), and 2.5 μg of each sample was treated with DNase (Ambion, Life Technologies). 3 ml of yeast culture was centrifuged, the pellet was to mechanical disruption with liquid nitrogen and was resuspended in buffer RLC (RNeasy Mini Kit) and further all procedures were done according to the manufacturer's protocol. The concentration and purity of the isolated RNA was determined using a UV spectrophotometer NanoDrop ND-1000 (Thermo Fisher Scientific, Wilmington, USA). The OD260/OD280 values of the RNA samples, reflecting their average purity, ranged between 1.9 and 2.1. Furthermore, integrity of the RNA isolates was verified through agarose gel electrophoresis, according to a standard method (Sambrook and Russell 2001).
The observed band patterns of all RNA samples exhibited sharp and intensive bands for 26S and 18S rRNA fractions (Figure S11).
2.11 Quantitative Real-Time Reverse Transcriptase Polymerase Chain Reaction (qRT-PCR)
cDNA templates for all quantitative real-time reverse transcriptase polymerase chain reactions (qRT-PCR) were synthesized from 2500 ng of total RNA using the MMLV RT kit (Evrogen, Russia) following the manufacturer’s protocol. RNA concentration was determined using a fluorometer (Qubit™ fluorometric instrument, Invitrogen Q32857, USA). Gene-specific primer for isoform phytase tPhyD-Bs (Fw: GAAGGACTGACAATCTATTATG, Rw: AACCGAGAACATCAATACC) were designed using Beacon Designer software and pair primers for the Act1 (Fw: CGAGCGAATGCACAAGGA, Rv: GCGGTGATCTTGACCTTGATG) were taken from the [“A new set of reference genes for comparative gene expression analyses in Yarrowia lipolytica”, Monika Borkowska, Wojciech Białas and Ewelina Celinska, FEMS Yeast Research, 20, 2020]. In all analysed samples, the value of the threshold cycle for AСT1 was the same (Figure S12) The qRT-PCR experiments were performed on an Eco Real-Time PCR System (Illumina, USA). Amplification efficiency for each primer set was calculated by serially diluting the exponential-phase cDNA template. Melting curve analysis was performed for each pair of primers after each run in an Eco Real-Time PCR System instrument to confirm the specificity of the primers. DNA contamination was checked by (no reverse transcription) PCR for each RNA sample in Eco Real-Time PCR System (Illumina, Inc., USA) using target primers (primer pair Pr-Phy-Bs/168/siamensis-f_BamHI and Pr-Phy-Bs/168/siamensis_NotI). No amplification was seen after 40 cycles of amplification (Figure S12). The specificity of each primer pair was verified via electrophoresis on a 1.2% agarose gel followed by ethidium bromide staining. The PCR products were directly purified from the reaction mixture using the Cleanup S-Cap kit (Evrogen, Russia) and sequenced using the Sanger method. Only one band was observed in the gel. All qRT-PCR experiments were performed on an Eco Real-Time PCR System (Illumina, USA). The amplification reaction was carried out using a ready-made mixture of PCR qPCRmix-HS SYBR+ROX (Evrogen, Russia) according to the manufacturer’s instructions. Amplification program: 5 min at 95 °C, 45 cycles (95 °C 15 s, 58 °C 20 s, 72 °C 30 s), and melting reaction product (Melting Curve program: 70–95 °C, +1 °C/s). The mRNA level was normalized to the expression of Act1 (Figure S13), which was constantly expressed under all experimental conditions. The qRT-PCR results were elaborated using the 2−ΔCT method [Livak, K.J.; Schmittgen, T.D. Analysis of relative gene expression data using real-time quantitative PCR and the 2(Delta Delta C (T)) method. Methods 2001, 25, 402–408]. All experiments were carried out in 3–4 replicates. Four independent experiments were conducted; the paper presents the results of the most representative experiment. Statistical analyses were performed using Microsoft Excel software. Statistical significance between experimental groups was determined using the Student’s t-test (confidence interval 95%) [Divisi, D.; Di, L.G.; Zaccagna, G.; Crisci, R. Basic statistics with Microsoft Excel: A review. J. Thorac. Dis. 2017, 9, 1734–1740.].
Round 2
Reviewer 2 Report
After revision, the manuscript has greatly improved and is now much clearer. All hypotheses are supported by scientific explanations. Supplementary materials have been added, which enhance the understanding of the manuscript and the data analysis. The quality is now high, and I consider it ready for publication in the Journal of Fungi.
Nothing to declare.
Author Response
We are grateful to you for a high evaluation for our study.